# Epigenetic landscape reveals MECOM as an endothelial lineage regulator

Jie Lv[1,7], Shu Meng[2,7], Qilin Gu ®[2,7], Rongbin Zheng[3,4], Xinlei Gao[1,2,3,4], Jun-dae Kim[2], Min Chen[3,4], Bo Xia[1], Yihan Zuo[3,4], Sen Zhu[1], Dongyu Zhao[1,2,3,4], Yanqiang Li ®[1,2,3,4], Guangyu Wang[1,2,3,4], Xin Wang[1,2,3,4], Qingshu Meng ®[5,6], Qi Cao ®[5,6], John P. Cooke ®[2] ✉, Longhou Fang ®[2] ✉, Kaifu Chen ®[1,2,3,4] ✉ & Lili Zhang ®[2,3,4] ✉

A comprehensive understanding of endothelial cell lineage specification will advance cardiovascular regenerative medicine. Recent studies found that unique epigenetic signatures preferentially regulate cell identity genes. We thus systematically investigate the epigenetic landscape of endothelial cell lineage and identify *MECOM* to be the leading candidate as an endothelial cell lineage regulator. Single-cell RNA-Seq analysis verifies that *MECOM*-positive cells are exclusively enriched in the cell cluster of bona fide endothelial cells derived from induced pluripotent stem cells. Our experiments demonstrate that MECOM depletion impairs human endothelial cell differentiation, functions, and Zebrafish angiogenesis. Through integrative analysis of Hi-C, DNase-Seq, ChIP-Seq, and RNA-Seq data, we find MECOM binds enhancers that form chromatin loops to regulate endothelial cell identity genes. Further, we identify and verify the *VEGF* signaling pathway to be a key target of MECOM. Our work provides important insights into epigenetic regulation of cell identity and uncovered MECOM as an endothelial cell lineage regulator.

The vascular endothelium forms the luminal lining of all blood and lymphatic vessels, and plays a dominant role in the regulation of vasomotion, vascular growth, and interaction of the vessel with circulating blood or lymphatic elements[1]. Dysregulated vascular development is associated with a myriad of congenital diseases or embryonic lethality[1,2]. During development, endothelial functions are essential for the viability of the embryo. Mesoderm-derived endothelial cells (EC) form the major blood conduits, from which angiogenesis proceeds. The process requires coordinated EC migration and proliferation, appropriate responses to environmental cues, and vascular remodeling. In the adult mammal, EC function is required for cardiovascular homeostasis as well as tissue remodeling or repair[1,2].

Our current understanding of the molecular programs governing EC lineage commitment and its cell-type specific functions remains limited. Aberrance of angiogenesis plays a role in age-related macular degeneration and the progression of atherosclerotic plaque and malignancy[3]. Alterations in EC phenotype, such as the increased expression of chemokines and inflammatory cytokines in response to metabolic or hemodynamic perturbations, are known to accelerate vascular diseases[4]. Accordingly, a greater understanding of the determinants of EC lineage will lead to improved therapeutic approaches for congenital and adult vascular diseases. Furthermore, such knowledge may facilitate development of therapy in regenerative medicine, such as

[1]Center for Bioinformatics and Computational Biology, Department of Cardiovascular Sciences, Houston Methodist Research Institute, Houston, TX, USA. [2]Center for Cardiovascular Regeneration, Department of Cardiovascular Sciences, Houston Methodist Research Institute, Houston, TX, USA. [3]Basic and Translational Research Division, Department of Cardiology, Boston Children's Hospital, Boston, MA 02115, USA. [4]Department of Pediatrics, Harvard Medical School, Boston, MA 02115, USA. [5]Department of Urology, Northwestern University Feinberg School of Medicine, Chicago, IL, USA. [6]Robert H. Lurie Comprehensive Cancer Center, Northwestern University Feinberg School of Medicine, Chicago, IL, USA. [7]These authors contributed equally: Jie Lv, Shu Meng, Qilin Gu. ✉e-mail: jpcooke@houstonmethodist.org; lhfang@houstonmethodist.org; kaifu.chen@childrens.harvard.edu; Lili.Zhang@childrens.harvard.edu

therapeutic transdifferentiation of fibroblasts to ECs to reperfuse ischemic tissue[5].

Genome-wide investigation of epigenetic information has emerged as a powerful tool to unveil molecular characteristics of distinct cell types[6–12]. We and several other groups discovered that across multiple cell types, cell identity genes display unique epigenetic signatures, e.g., broad H3K4me3[13,14] and super-enhancers as marked by broad H3K4me1 or broad H3K27ac[15–17], whereas most other active genes display a sharp enrichment pattern (narrow but high enrichment peak) of these modifications[13,14].

In this study, to define EC identity genes, we systematically analyze the epigenetic landscape of ECs for the discovery of regulators of EC identity. We further analyze epigenome and transcriptome to characterize the role of the transcription factor *MECOM* in regulating EC transcription programs. Finally, we reveal a key EC role of *MECOM* in the transcriptional activation of *VEGFR2*, which is the receptor protein in the VEGF signaling pathway.

## Results

### Broad H3K4me3 and super-enhancers co-exist at super active chromatin domains in ECs

To systematically investigate the epigenetic landscapes of ECs, we analyzed 22 genome-wide chromatin marks in human umbilical vein endothelial cells (HUVEC). These included ChIP-Seq data of 11 distinct histone modifications, DNase-Seq data that indicate chromatin openness, and ChIP-Seq data of 10 chromatin-binding proteins. Since a broad pattern of H3K4me3 enrichment (Broad H3K4me3) was reported to mark cell identity genes[13,14], we investigated the breadth of H3K4me3 as well as the other 21 chromatin marks in HUVECs (Fig. s1a). We observed that the breadth of these different chromatin marks is strongly correlated. The strongest positive correlations are among five activating histone modifications (Fig. s1b, Supplementary Data 1). Intriguingly, three of them have been reported in other cell types to mark cell identity genes when they are broad at the gene loci. These include broad H3K4me3 that covers both the promoter and gene body of cell identity genes[13,14], and broad H3K4me1 and broad H3K27ac that denote super-enhancers for cell identity genes[15–17]. The other two of these five modifications include H3K9ac and H3K4me2. Genes with broad enrichment of activating histone modifications, e.g., broad H3K9ac (Fig. s1c,d), show broad chromatin openness, as indicated by broad enrichment of DNase-Seq signal (Fig. s1e). Further, these genes are enriched with ChIP-Seq signals of RNA polymerase II (Pol II) across their gene bodies (Fig. s 1f), indicating a high frequency of transcription elongation. In general, these genes display higher expression when compared to genes associated with sharp or random enrichment peaks of activating histone modification (Fig. s1g). These patterns observed for H3K9ac were also verified when we repeated the same analysis using the other activating histone modifications (Fig. S2a–d). These results suggest that the super-enhancer, broad H3K4me3, and broad enrichment of other activating marks can occur in the same genomic regions. These regions appeared to be super active chromatin domains, in which the genes showed increased transcription when compared to genes in other regions.

### EC identity genes are enriched in super active chromatin domains

To investigate the association between super active domains and cell identity genes in ECs, we defined a set of 206 positive EC regulators as the intersection between the genes in the GO pathways that each have the keyword endothelial in the pathway name and the genes upregulated in EC compared to ESC (Supplementary Data 2). The broad enrichment of each of 5 histone modifications, which are the 5 modifications whose widths showed the strongest correlations with each other (Fig. s1b), displayed exceptionally strong associations with these positive EC regulators (Fig. 1a, S2e, f). For each histone modification,

we analyzed the overlap between these positive EC regulators and top 1000 genes ranked by the breadth of the modification. The *P* values of the overlap, as determined by the Fisher's exact test, range between 1.3e−53 and 7.4e−43 (Fig. 1a). Broad chromatin openness, as indicated by the broad enrichment of DNase-Seq signal, also exhibits strong association with these positive EC regulators (*P* value of 6.2e−45) (Fig. 1a). The four other activating modifications analyzed in this study, including H3K79me2, H3K9me1, H4K20me1 and H3K36me3, also show similar association with these positive EC regulators, but with less significant *P* values ranging between 1.2e−22 and 5.8e−09 (Fig. 1a). This result was further confirmed by analyzing other enrichment scores such as the fold of enrichment (Fig. S2e) and the number of overlapped genes (Fig. S2f). This result suggests that activating histone modifications are heterogeneous in their association with these positive EC regulators. The association was not detected for broad enrichment of the repressive histone modification H3K27me3 or H3K9me3. Therefore, the epigenetic signatures previously reported for cell identity genes in cell types other than ECs, including the broad H3K4me3 and super-enhancer (broad H3K4me1 or broad H3K27ac), were successfully confirmed to be indicators of EC positive regulators by our analysis in ECs. Further, our systematic analysis in ECs indicated that the broad chromatin openness and broad enrichment of H3K9ac and H3K4me2 are comparable or superior to previously described epigenetic signatures, e.g., broad H3K4me3 and super-enhancer, in enriching positive EC regulators.

We next combined the activating histone modifications for identification of EC identity genes. We ranked genes based on width of their associated enrichment peaks for each of the five modifications whose broad enrichments showed the strongest association with positive EC regulators. We then took the average of the 5 rank values for each gene to generate a single rank list. GO analysis of the top 1000 genes in this rank revealed enrichment in important EC-related pathways, including VEGF signaling pathway (GO:0048010), EC differentiation (GO GO:0045446), and EC migration (GO:0043536) (Fig. 1b). We next evaluated the importance of these 1000 top-ranked genes in EC gene regulation network constructed using the CellNet algorithm (See Gene Regulation Network analysis in Supplemental Method). A hub gene that serves as master regulator in a cell identity gene regulation network is expected to be connected by a large number of network edges to many other genes, which are adjacent upstream regulators or downstream targets of the hub gene in the network[18]. When compared to genes ranked randomly, ranked based on high expression level in ECs, or ranked based on up- or down-regulation in ECs relative to ESCs, the top genes ranked in our method, i.e., ranked by the average rank of widths of the five histone modifications, manifest significantly more edges in an EC gene regulation network (Fig. 1c) and encode a larger proportion of transcription factors (Fig. 1d). These data suggest a feasibility to utilize epigenetic landscape for identification of EC identity regulators.

### Epigenetic landscapes reveal *MECOM* as a lineage regulator for ECs

We next decided to focus on the transcription factor *MECOM*, which we find to show strong epigenetic signature of identity genes in ECs and was also identified as an EC identity gene in our recent machine learning analysis of EC epigenetic landscape[18]. More recently, another group demonstrated that *MECOM* KD in hESC-derived EC upregulates non-arterial EC markers[19]. The EC *MECOM* locus is marked with a broad H3K4me3 as well as a super-enhancer with broad H3K4me1 and broad H3K27ac (Fig. 1e). In addition, the *MECOM* promoter in ESCs is a bivalent domain as indicated by a cooccurrence of both H3K27me3 and H3K4me3 marks in ESCs (Fig. 1e). Notably, a bivalent domain, when appears in ESCs, was known to mark somatic cell lineage factors[20]. The H3K27me3 at ESCs diminished in ECs, whereas the H3K4me3 domain is lengthened in ECs relative to ESCs. Consistently,

other activating histone modifications, i.e., H3K9ac, H3K4me2, H3K36me3 and H3K79me2, also become broader at *MECOM* in ECs than in ESCs (Fig. S3a). Expression level of *MECOM* increased significantly during EC differentiation from ESCs (Fig. 1f). Subsequently, we studied transit heterokaryon formation between human ECs and mouse ESCs. The generation of such heterokaryons induces the expression of canonical mouse EC genes, and is an approach employed to study early events during EC reprogramming[21]. We analyzed *MECOM* expression during transient heterokaryon formation between human ECs and mouse ESCs based on an RNA-Seq dataset[21]. We observed that *MECOM* expression was significantly increased at 6 h after fusion but

was decreased at 12 h and 24 h (Fig. 1g). To further study the role of *MECOM* in EC lineage, we assessed its expression during fibroblast-to-EC transdifferentiation[22]. In this model, *MECOM* expression was increased (Fig. 1h). *MECOM* displayed broadening of activating marks H3K4me3 and H3K27ac in HUVECs compared to fibroblast NHLFs (Normal Human Lung Fibroblasts) (Fig. 1e). Furthermore, among the 71 transcription factors in the gene regulation network defined for ECs by the algorithm CellNet, *MECOM* is connected by the largest number of network edges (Fig. 1i, Supplementary Data 3). Interestingly, 7 out of the top 10 transcription factors with the largest number of network edges have been reported to play critical roles in the transcriptional

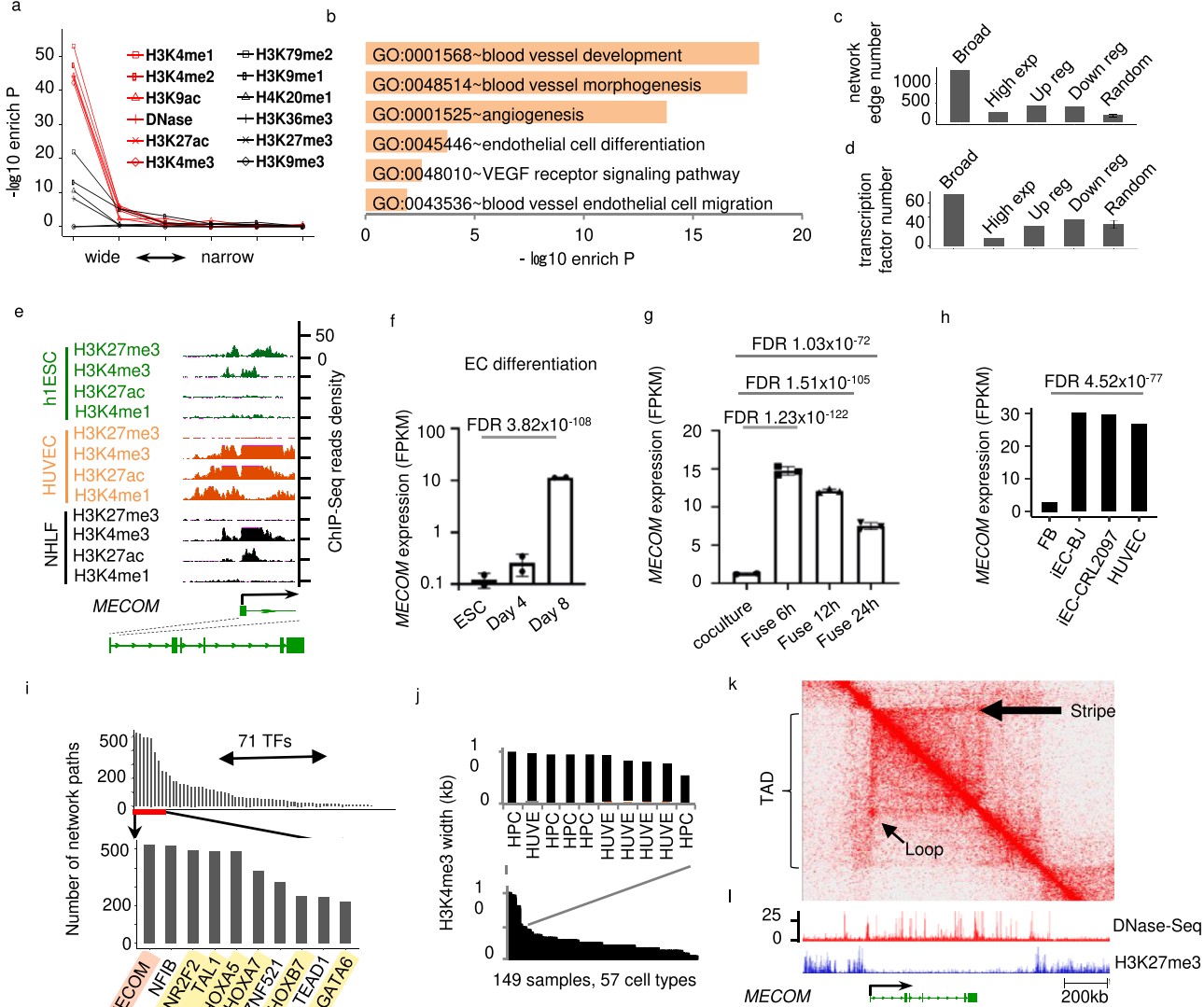

**Fig. 1 | Epigenetic landscapes suggest MECOM as a lineage regulator for Endothelial cell (EC). a** Enrichment of EC identity regulators plotted against the width of enrichment peak for each chromatin mark. **b** Pathways enriched in EC putative identity regulators defined by broad enrichment of activating histone modifications. **c** Number of regulation network edges connected to and (**d**) Number of transcription factors in EC putative identity regulators or other gene groups, of which each has the same number of genes. **e** Signal density of individual chromatin marks at the *MECOM* locus in h1ESC, HUVEC and NHLF. **f–h** Expression level of *MECOM* at individual intervals during EC differentiation mESC (**f**), the formation of mESC-hEC heterokaryon (**g**), and human fibroblast to EC transdifferentiation (**h**). **i** Number of gene regulation network paths connected by each transcription factor (TFs). MECOM is highlighted in red and known EC regulators are highlighted in yellow. Gene regulation network for EC was defined by algorithm and database of

the software CellNet. **j** H3K4me3 peak width at *MECOM* locus in individual ChIP-Seq samples. Sample information is provided in Supplementary Data 4. **k** Two-dimensional heat map showing the chromosomal contact frequency in HUVEC around *MECOM* locus. **l** Signal density of DNase-Seq and H3K27me3 ChIP-Seq in HUVEC around MECOM locus. Broad: broad active histone modifications; High exp: high expression; Up reg or Down reg: up or downregulated in EC relative to ESC. Random: randomly selected genes. Data are presented as mean values ± SD (**f**, **g**). $n = 2$ biologically independent samples (**f**), $n = 3$ biologically independent samples (**g**). *P* values are determined by two-tailed Fisher Exact test implemented in R v4.0.2 (**a**), modified two-tailed Fisher Exact test implemented in DAVID v6.8 (**b**), two-tailed Negative Binomial test (**f**, **g**), and two-tailed Poisson test implemented in edgeR v3.14.0 (**h**). EC data were from HUVECs (See Supplementary Data 4).

control of EC differentiation, development, and function, including *NR2F2*[23], *TAL1*[24], *HOX* gene family[25,26] and *GATA6*[27]. For instance, the *NR2F2* is a key lineage factor for vein EC identity. Depletion of NR2F2 in vein ECs was found to cause a transition toward the arterial EC identity, whereas forced expression of NR2F2 in arterial ECs caused a transition toward the vein EC fate[2]. Taken together, these EC-specific epigenetic and expression data predict that *MECOM* is a putative EC lineage regulator.

To gain more information regarding super active domains at *MECOM*, we further analyzed the H3K4me3 signal for *MECOM* in 149 ChIP-seq datasets derived from 57 cell types (Supplementary Data 4). HUVEC is one of the two cell types showing the broadest H3K4me3 at the *MECOM* locus (Fig. 1j). Specifically, 5 out of 10 samples that display the broadest H3k4me3 at *MECOM* come from HUVECs, with the other 5 samples derived from hematopoietic stem and progenitor cell (HSPC) (Fig. 1j). DNase-Seq analysis further showed that the *MECOM* gene promoter had the strongest chromatin openness in EC, including both HUVEC and HPAEC, whereas *MECOM* in HPC was associated with the second strongest chromatin openness compared to other cell types (Fig. S3b). *MECOM* mRNA abundance is also the highest in five types of EC when compared to HPC and other immune cells (Fig. S3c). Intriguingly, endothelium and HPC fate are associated with early development, and *MECOM* is required for the formation of HPC but inhibits the differentiation of HPC toward downstream lineages[28]. These results further verified the reliability of applying broad H3K4me3 as a signature to define a cell identity gene. We also analyzed DNase-Seq signal to compare chromatin openness across 4 ESC, 6 EC, and 5 fibroblast samples; we found ECs consistently manifested broader open chromatin regions at the *MECOM* locus (Fig. S3d). This result implies that the chromatin region surrounding *MECOM* is selectively activated for EC specification.

We next investigated the epigenetic landscape of *MECOM* in diverse EC subtypes. By comparing the DNase-Seq signal in the gene locus for *MECOM*, we found a large cluster of enrichment peaks in the 4 Blood Vessel Endothelial Cell (BEC) subtypes, with some of these peaks diminished in the 2 Lymphatic Vessel Endothelial Cell (LEC) subtypes (Fig. S3e, f). Consistent with DNase-Seq data, *MECOM* shows high expression in BEC but moderate expression in LEC (Fig. S3g). Therefore, *MECOM* expression may be fine-tuned in distinct EC subtypes by modulating chromatin openness at individual sites within a super active domain.

### *MECOM* resides within a single topologically associating domain (TAD) in ECs

Enhancers may interact with target genes via chromatin looping to regulate transcription. To investigate the interaction between *MECOM* locus and its associated enhancers, we analyzed Hi-C data at the *MECOM* locus in HUVECs. We found that *MECOM* is located inside a single topologically associated domain (TAD) (Fig. 1k), in which the boundaries are at the two ends of *MECOM*. This result suggests that the regulation of *MECOM* expression might be autonomous. A strong stripe in the interaction heat map indicates frequent interaction between the promoter and the gene body region of *MECOM*, which is consistent with recent discovery that active promoters can maintain contact with the gene body during transcription and beyond[29]. Further, the strongest interaction occurs between the two ends of the stripe, suggesting a chromatin loop whose anchors display frequent interaction with the entire chromatin loop that covers the entire gene body of *MECOM* (Fig. 1k). The TAD at *MECOM* also exists in H1 ESCs, agreeing with previous knowledge that TADs tend to be conserved between cell types (Fig. S3h). However, the stripe and loop observed in HUVECs appeared to be weaker (if not absent) in H1 ESCs (Fig. S3h). In contrast, we observed that in a nearby genomic region, multiple loops were present in H1ESCs but absent in HUVECs (Fig. S3h). In agreement with these observations, the gene body region of *MECOM* displays a

large cluster of DNase-Seq enrichment peaks (Fig. 1l), which often indicate active enhancers bound by transcription factors. The whole TAD domain is depleted of H3K27me3, a repressive histone modification, whereas the genomic regions flanking the TAD domain are broadly enriched with H3K27me3 and are depleted of DNase-Seq signal (Fig. 1l). These data indicate that *MECOM* is localized in a super active domain that forms a single TAD in HUVECs.

### Single-cell RNA-seq analysis reveals *MECOM* as a molecular signature of iPSC-derived ECs

We further utilized single-cell RNA-Seq data[30] to investigate the dynamic change of *MECOM*-positive cell population during in vitro EC differentiation. We analyzed single-cell RNA-Seq data on days 8 and 12 of EC differentiation from iPSCs. We defined 11 individual clusters from cells pooled from both dates (Fig. 2a). The expression of *MECOM* is not random across these subpopulations, with cells expressing *MECOM* nearly constrained within a single subpopulation (cluster 0) (Fig. 2b). The same cell cluster also displays exclusive expression of *VEGFR2* (Fig. 2b), a key receptor in the Vascular Endothelial Growth Factor (VEGF) Signaling Pathway, as well as other EC markers, e.g., the *CDH5* (*CD144*, Fig. S4a), *VWF* and *ERG*, and thus is identified as bona fide iPSC-derived ECs (Fig. 2c, Fig. S4a). Further, the *MECOM*-positive cell cluster expanded from day 8 to day 12 by over two-fold (Fig. 2d). These results provided single-cell evidence that *MECOM* up-regulation is a molecular signature of EC differentiating from iPSC. As MECOM also has a reported role in HPC, we further checked the expression level of HPC markers *ITGA2B* and *Runx1*. Cluster 0 cells didn't show enrichment of those HPC markers (Fig. S4b, c). Cluster 0 cells further expressed an overall significantly higher level of the 206 EC identity genes (Supplementary Data 2) when compared to 103 HPC differentiation genes (GO:0002244) (Fig. S4d). These results indicated that the cells in Cluster 0 are induced EC cells rather than HPC. The correlation between *MECOM* expression and EC marker genes was further observed in another single-cell RNA-Seq dataset[31], which were generated during EC differentiation (Fig. S4e). We further carried out qPCR to detect gene expression at early stage of iPSC-EC differentiation from day 0 to day 6. We found that *MECOM* started expression on day 2. *MECOM* expression upregulation was greater than four tested EC markers (*CDH5*, *KDR*, *DLL4*, and *EFNB2*) on days 2 and 3, whereas the upregulation of *CDH5* becomes greater after days 4 (Fig. S4f). This result suggested that *MECOM* expression preceded the expression of these known EC marker genes in the differentiation process.

### MECOM is required for differentiation from iPSC to EC lineage

To verify the role of MECOM in EC lineage regulation, we generated *MECOM* knockout (KO) iPSC single colony (Fig. S4g, h) and evaluated the effect of the knockout on EC differentiation (Fig. S4i). The depletion of *MECOM* RNA expression level was associated with a marked reduction in the yield of ECs during differentiation, from 9.29% to 0.44% (Fig. 2e–g). Also, the decreased efficiency of EC differentiation was associated with reduced expression of EC markers, i. e., *eNOS* (Fig. 2h), and *VWF* (Fig. 2i), as well as higher residual expression of iPSC markers after differentiation, i. e., *OCT4* (Fig. 2j) and *SOX2* (Fig. 2k). We further confirmed these results by knocking down *MECOM* using siRNA in human iPSC (Fig. S5a–f). *MECOM* knockdown reduced the differentiation efficiency of iPSCs to ECs from 20.9% to 11.2% (Fig. S5a, b), which is consistent with a 50% reduction in *MECOM* RNA expression level in the iPSC (Fig. S5c). Further, *MECOM* expression is reduced by only 20% in the induced ECs with *MECOM* KD (Fig. S5d), suggesting that these induced ECs might have been derived from the iPSCs in which *MECOM* is partially knocked down. The decreased efficiency of EC differentiation from *MECOM* KD iPSC (iPSC-EC) is further supported by the residual expression of iPSC marker *OCT4* (Fig. S5e), as well as lower expression of EC markers *CD31* (Fig. S5f). To verify if the decreased efficiency of EC differentiation was caused by impaired iPSC

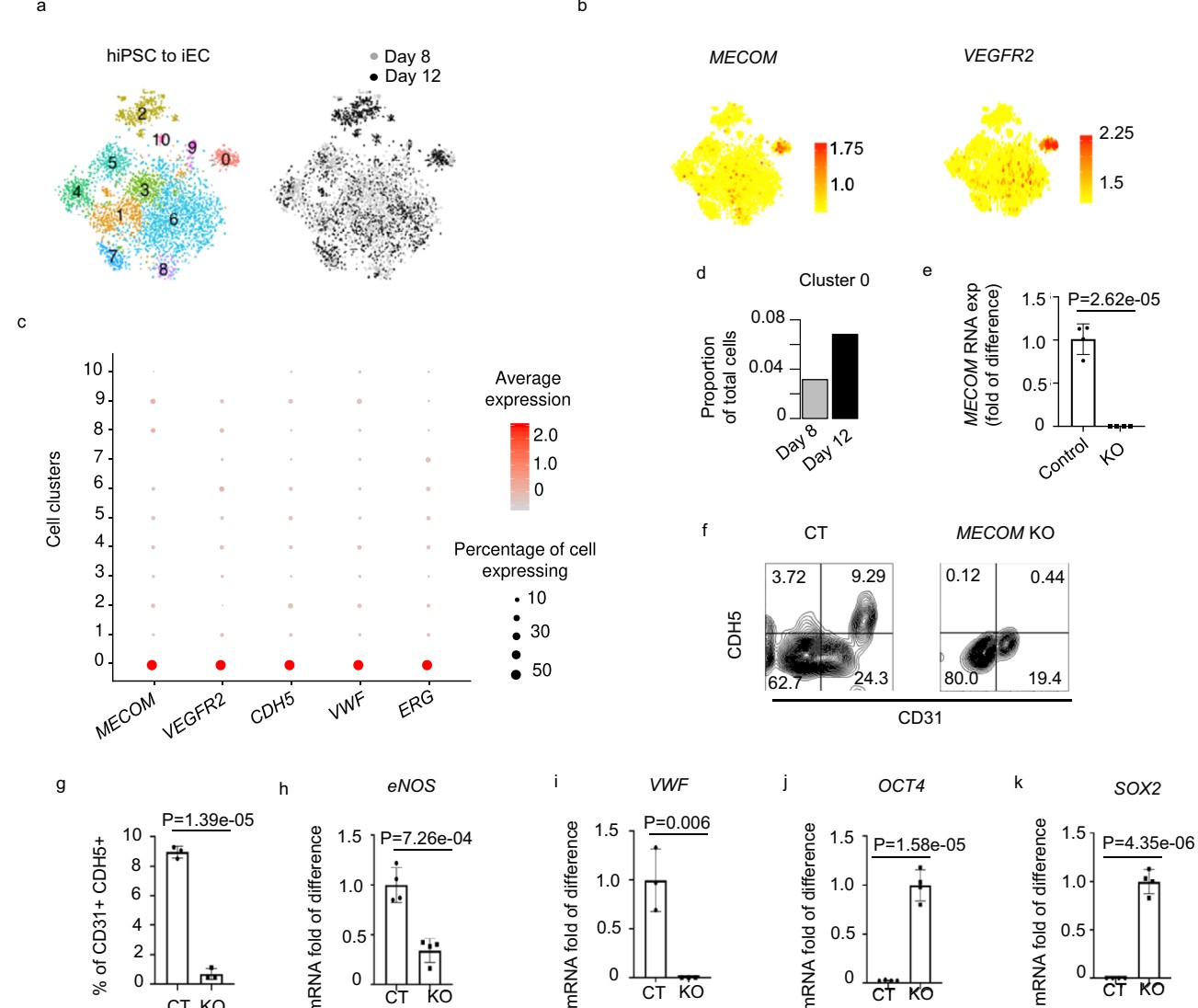

**Fig. 2 | MECOM is required for EC differentiation from hiPSC. a–d** scRNA-seq analysis reveals *MECOM* as a molecular signature of iPSC-derived ECs. (**a**) T-SNE plot of combined cells from days 8 and 12 of EC differentiation. 11 cell clusters are defined based on transcriptomic similarity. **b,c** Expression profiles of *MECOM* and *VEGFR2* across 11 individual clusters (**b**) compared with other known EC markers (*CDH5*, *VWF*, *ERG*) (**c**). **d** Proportion of cells assigned to the *MECOM* positive cluster 0 on day 8 and day 12. **e–k** Experimental verification of MECOM as a lineage factor required for EC lineage transition from iPSCs (**e**) Relative expression level of *MECOM* in *MECOM* KO sample. **f** FACS analysis of CD31⁺CDH5⁺ ECs. **g** Quantification of CD31⁺CDH5⁺ EC percentage. **h–k** Relative gene expression levels of *eNOS* (**h**), *VWF* (**i**), *OCT4* (**j**), and *SOX2* (**k**) in *MECOM* KO sample. Data are presented as mean values ± SD (**e**, **g–k**). *n* = 4 biologically independent samples (**e–k**), *P* values are determined by two-tailed Student's *t* test. Source data are provided as a Source Data file.

pluripotency after *MECOM* KO, we performed qPCR of iPSC pluripotency markers in WT and *MECOM* KO iPSC. We found that *MECOM* KO did not decrease the expression of well-known pluripotency markers in iPSC, including the *OCT4*, *SOX2*, *NANOG*, *TERT*, *ZFP42*, and *UTF1* (Fig. S5g). We instead observed either significant or slight up regulation of these marker genes in iPSCs upon *MECOM* KO. We further performed Alkaline phosphatase (AP) staining, a key method to assess the pluripotency of iPSC. We found that the AP activity level didn't show significant differences between WT and *MECOM* KO iPSC (Fig. S5h, i). Taken together, our data indicated that *MECOM* is required for EC lineage transition from iPSCs. *MECOM* depletion significantly reduced the yield of ECs after differentiation, without impairing iPSC pluripotency.

**MECOM is required to maintain EC phenotypes and functions**
To further test the role of MECOM in the maintenance of EC phenotype and function, we disrupted *MECOM* in HUVECs using CRISPR-Cas9 to

generate *MECOM* KO cell pool (not a single clone, Fig. 3a). *MECOM* disruption significantly reduced *CDH5* and *VEGFR2* mRNA level in HUVEC cells (Fig. S6a). Gene Set Enrichment Analysis (GSEA) further showed that the 206 reported EC positive regulators (Supplementary Data 2) are significantly enriched with down-regulated genes upon *MECOM* depletion (Fig. S6b), suggesting loss of EC identity upon *MECOM* depletion. *MECOM* disruption significantly impaired angiogenesis, as indicated by reduced tube formation in vitro (Fig. 3b). Specifically, both the number of branching points (Fig. 3c) and tube length (Fig. 3d) were significantly decreased. In addition, *MECOM* disruption also impeded HUVEC cell migration, as evidenced by the lower rate of wound closure in *MECOM* KO HUVEC pool using the wound scratch assay (Fig. 3e). The distance between the two edges of the wound line is significantly larger in *MECOM* KO HUVEC pool (Fig. 3f). The LDL uptake by HUVECs was decreased by four folds in response to *MECOM* KO (Fig. 3g). The NO production (Fig. 3h) of HUVECs was also attenuated by the *MECOM* KO. Further, HUVEC cell proliferation was

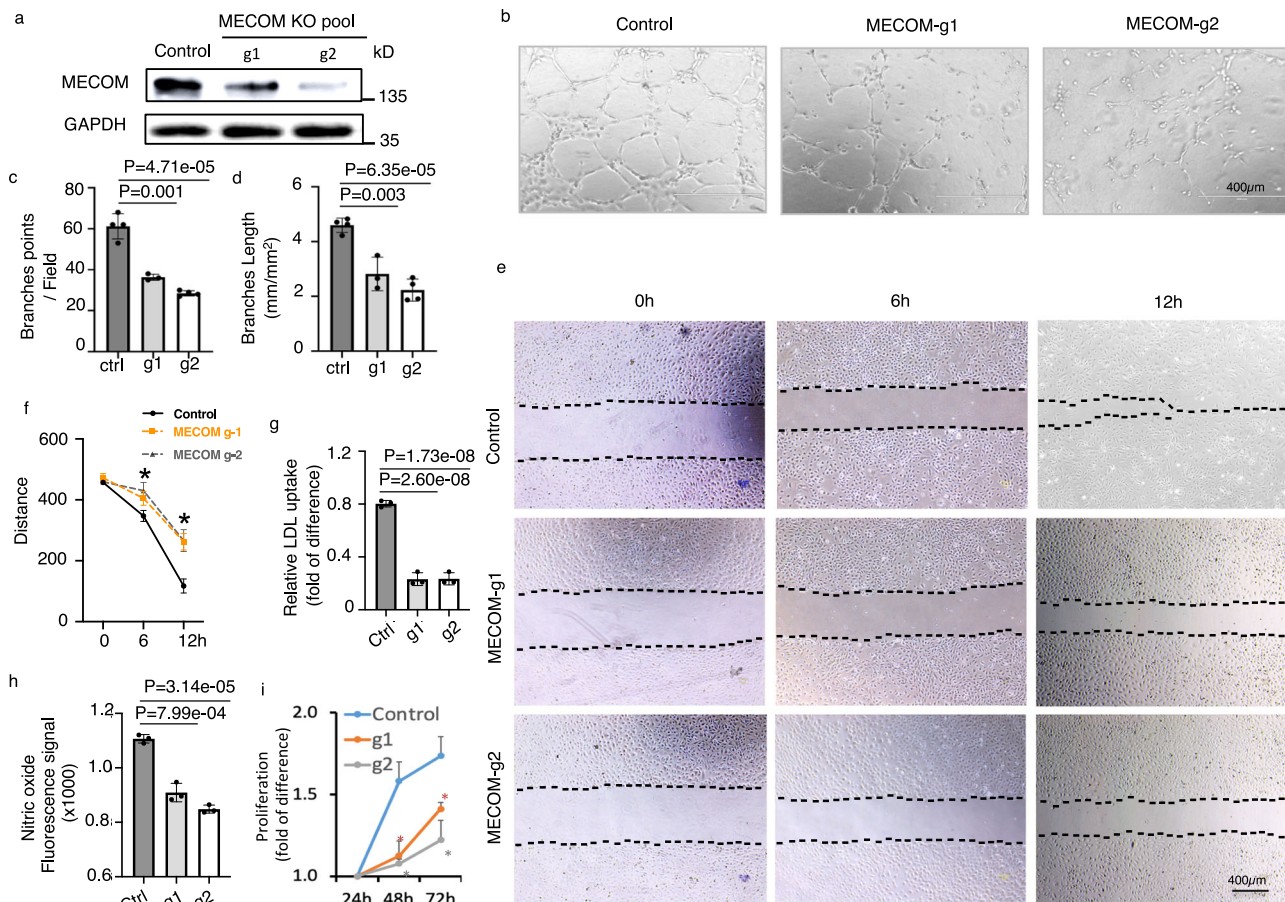

**Fig. 3 | MECOM is required for HUVEC phenotype and function. a** Western blot to show MECOM protein levels. **b** Representative images of in vitro tube formation in control and MECOM-KO HUVECs pool. **c** The number of EC tube branching points per high power field. **d** The length of EC branching tube. **e** Representative wound closure images at 0 h, 6 h, and 12 h after scratch. **f** Distance between the two borders (dotted lines in **e**) in wound closure images. **g**, **h** LDL uptake (**g**) and Nitric oxide production (**h**) in control and MECOM-KO HUVECs pool. **i** HUVEC Proliferation. Two CRISPR-Cas9 gRNAs g1 and g2 were tested. Error bars represent variation between replicates; $n = 3$ biologically independent samples. Data are presented as mean values ± SD; *$P < 0.05$, $P$ values determined by two-tailed Student's $t$ test. Source data are provided as a Source Data file.

significantly decreased by *MECOM* KO (Fig. 3i). To verify if the other effects observed in *MECOM*-depleted HUVECs were indirectly induced by the defective cell proliferation or cell viability, we treated the HUVEC with Mitomycin C (Fig. S7a), which has been reported to inhibit cell proliferation[32]. Consistent with the previous results, *MECOM* depletion still significantly impaired EC functions and thus, these other effects is not simply caused by defective cell proliferation and viability (Fig. S7b–i). Collectively, these data highlight the important role of *MECOM* in maintaining EC functions and phenotypes.

### *mecom* depletion impairs zebrafish angiogenesis
To explore the role of MECOM in ECs in vivo, we generated *mecom* KO zebrafish. We targeted a specific sequence at the 3rd exon of zebrafish *mecom* (Fig. 4a) and validated the *mecom* KO by sequencing (Fig. 4b, c), qPCR (Fig. 4d), and western blot (Fig. 4e). In contrast to the control zebrafish, the zebrafish with *mecom* loss-of-function mutation (*fli1:egfp; mecom$^{-/-}$*) displayed impaired angiogenesis when compared to the control fish (*fli1:egfp*) (Fig. 4f). Both the length of the intersegmental vessels (ISV) (Fig. 4g) and the subintestinal veins (SIV) (Fig. 4h) were significantly reduced in the *mecom* KO zebrafish. We also found diminished SIV sprouting in the *mecom* KO zebrafish (Fig. 4i). In addition, we further validated these effects of *mecom* depletion on embryonic angiogenesis by knocking down *mecom* using Morpholino antisense oligos (Fig. S8). Consistent with previous report[33,34], *mecom* KO or knock down didn't affect the establishment of dorsal aorta or the major vein, a process known as vasculogenesis. Instead, our data showed that *mecom* deficiency impairs angiogenesis, which includes the establishment of ISV and SIV.

### MECOM regulates the transcriptional program for EC differentiation and function
To understand the mechanism by which MECOM regulates EC identity, we performed RNA-Seq for genome-wide expression profiling in the control and *MECOM* KO HUVEC pool. We found that *MECOM* KO downregulated 2476 and upregulated 2733 genes (Fig. 5a). The downregulated genes were enriched in EC-related pathways, e.g., EC differentiation (GO:0045446), VEGFR signaling pathway (GO:0048010), and Regulation of EC proliferation (GO:0001936). Up-regulated genes were enriched in pathways associated with cell cycle phase (GO:0022403), cell division (GO:0051301), and mitosis (GO:0007067) (Fig. 5b). The down- and up- regulated genes in ECs overlap significantly with those up and down regulated in ECs compared to ESCs, respectively (Fig. 5c). These results suggest that MECOM plays its role in HUVECs by regulating pathways associated with EC lineage specification and EC functions.

Many cell identity genes were known to play an oncogenic role in cancer, e.g., ETV2 induces EC lineage when expressed in fibroblast[35], but plays an oncogenic role in tumorigenesis[36]. Similarly, dysregulated MECOM was known to play an oncogenic role, e.g., in ovarian cancer[37,38]. Intriguingly, we found only 643 genes (365 down and 278

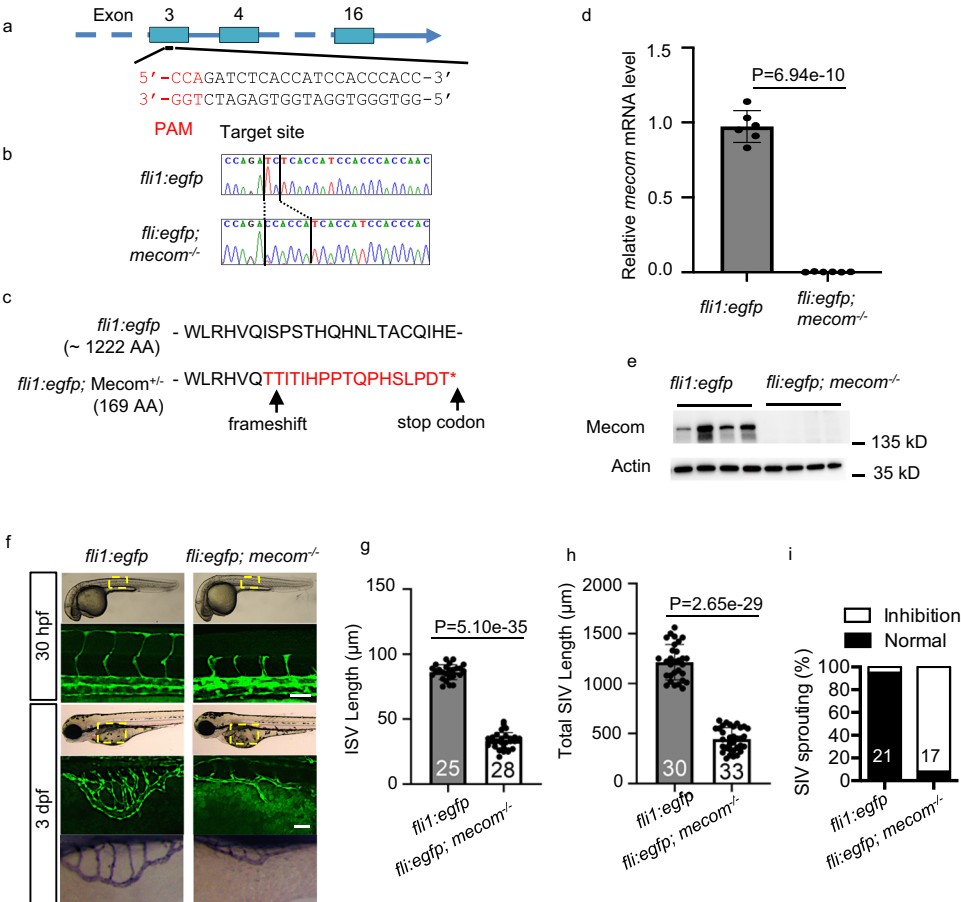

**Fig. 4 | Depletion of *mecom* impairs zebrafish blood vessel formation. a** CRISPR-cas9 target site on *mecom* gene. The target sequence is based on the anti-sense strand. **b** DNA sequencing of *mecom* mutant animals. The PCR product covering the designed mutation site was cloned into T-vector and the individual colony harbored the insert was sent for DNA sequencing. The TC → CCACCA indel causing frameshift in mecom gene is highlighted. The vertical bars demarcate the mutation site. **c** *mecom* gene disruption produces a truncated protein. **d** Relative gene expression levels of *mecom*. **e** Western blot to show Mecom protein levels. **f** Confocal imaging of intersegmental vessel (ISV) at 30 hpf (top) and subintestinal vein (SIV) at 3 dpf (middle) as well as alkaline phosphatase staining of SIV at 3 dpf (bottom) of *mecom* mutant and wild type control. **g, h** Quantification of ISV length (**g**) and SIV length (**h**). **i** Percentage of larvae showing normal or inhibited SIV sprouting. Scale bars, 100 μm. Data are presented as mean values ± SD; *n* = 6 (**d**) and *n* = 4 (**e**) biologically independent samples, numbers of biologically independent animals analyzed are indicated in the bars of (**f**–**i**). *P* values are determined by two-tailed Student's *t* test. Source data are provided as a Source Data file.

up) regulated in response to *MECOM* knockdown in ovarian cancer cell SKOV3 (Fig. 5a). This number was eight-fold smaller than the number of genes regulated by *MECOM* in ECs, and ten-fold smaller than the number of differential genes between ECs and ESCs (Fig. 5a). To account for technical effects such as potential inconsistency of knockdown efficiency between independent experiments, we next used the same number (2000) of top differential genes from each of these analyses to analyze the overlap between these gene groups (Fig. 5c). Although there is significant overlap of genes regulated by *MECOM* in ECs and in SKOV3 cells, a three-fold greater overlap is observed between the genes regulated by *MECOM* in ECs and the differential genes in ECs relative to ESCs. Therefore, the transcription program regulated by *MECOM* showed a stronger association with the EC program than with the oncogenic program.

**MECOM binds enhancers to regulate genes in EC pathways**

MECOM encodes a transcription factor that was known to directly bind DNA for activation or repression of target genes[39]. Although a transcription factor may bind many different enhancers across the genome, we speculate that the DNA motif recognized by the transcription factor will be similar in many of these enhancers. We utilized public ChIP-Seq data for MECOM in the cell line SKOV3[40] to define MECOM binding sites, and next defined the consensus binding motif in these sites (Fig. 5d top panel). As expected, the motif is enriched at the center of MECOM binding sites (Fig. 5d bottom panel). The MECOM binding motif is significantly enriched in both proximal and distal regions around transcription starting sites (TSS) of genes that were up or down regulated in ECs when compared to ESCs (Fig. 5e top panel) and the genes that were down or upregulated in ECs in response to *MECOM* KO (Fig. 5e bottom panel). Pathway analysis indicated that MECOM binding motif is enriched in promoter (from TSS to 5 kb upstream) of genes in EC-related pathways such as the Endothelial Differentiation (GO:0045446) and the VEGF signaling pathway (GO:0048010) (Fig. 5f). We therefore hypothesized that the regulatory elements bound by MECOM are more active in ECs compared to ESCs. This prompted us to compare the enrichment of the H3K27ac signal, a known mark for active enhancer, at MECOM target sites (chromatin regions harboring MECOM motif) between HUVECs and ESCs. Aggregate analysis showed that MECOM target sites in HUVECs showed two-fold higher H3K27ac enrichment (Fig. 5g top panel), suggesting that MECOM target sites are more active in HUVECs than in ESCs. The pattern of higher H3K27ac enrichment in HUVECs is also prominent at the level of individual MECOM target sites (Fig. 5g bottom panel). Taken together, our data suggest that MECOM binds active enhancers (either directly or indirectly to the inferred motif

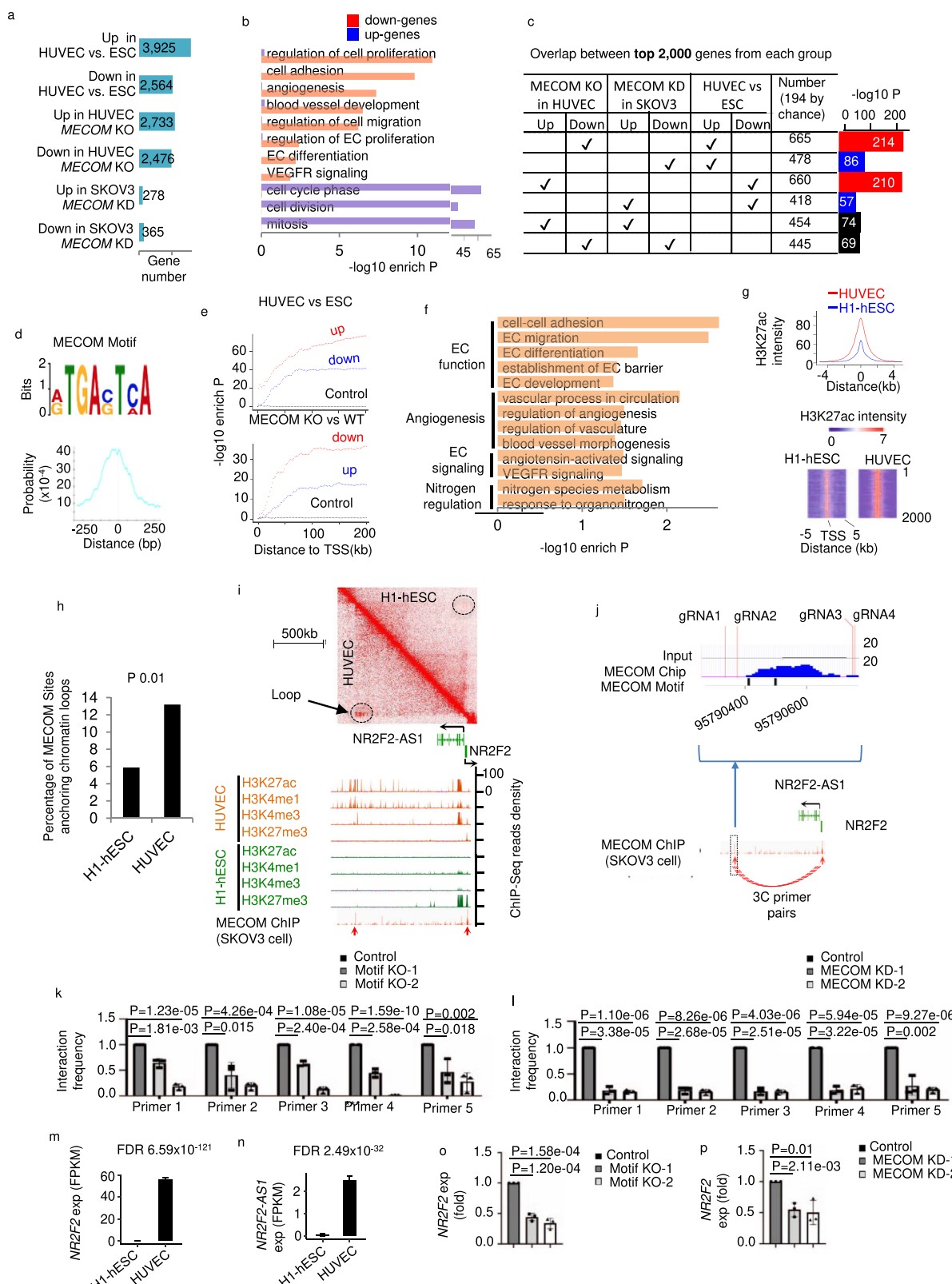

regions), to regulate the transcriptional program for EC lineage establishment and functions.

## MECOM target sites formed chromatin loops to regulate target genes in ECs

We speculate that the binding of MECOM on enhancers to regulate the transcription of target genes will trigger chromatin interactions between the enhancers and target genes. Indeed, Hi-C analysis indicated that MECOM binding motifs in the genome more frequently participate in forming chromatin loops in HUVECs than in ESCs (Fig. 5h). Specifically, 13.20% of binding motifs for MECOM overlap with genomic sites connected by strong chromatin loops in ECs, compared to 5.89% in ESCs. For example, we detected two major target sites for MECOM near the gene *NR2F2* (Fig. 5i), which is annotated to

**Fig. 5 | MECOM binds enhancers to regulate the transcriptional program for EC differentiation and function. a** Number of differentially regulated genes comparing pairs of conditions. **b** Pathways significantly enriched in differential genes between WT and *MECOM* KO HUVEC pool. **c** Pairwise comparisons among genes regulated by *MECOM* in HUVEC, genes regulated by *MECOM* in SKOV3 cell, and genes regulated in EC differentiation. **d** The MECOM binding motif (top) and its distribution around centers of ChIP-Seq enrichment peaks (bottom). **e** Enrichment of MECOM binding motif plotted against distance to genes differentially expressed in EC differentiation (top) or upon MECOM KO in EC (bottom). **f** Pathways enriched in genes associated with MECOM motif in promoter region from TSS to 5 kb upstream. **g** Average signal intensity of H3K27ac at MECOM target sites in HUVEC and H1-hESC (top) and heat map to show the signal intensity of H3K27ac at each base pair (column) of individual MECOM target site (row) (bottom). **h** Percentage of MECOM target sites overlap with chromatin loops in HUVEC and H1-hESC

respectively. **i** Two-dimensional heat map of chromosomal contact frequency (top), signal density of chromatin marks and MECOM ChIP-Seq (bottom) at *NR2F2/NR2F2-AS1* locus in HUVEC. Red arrows indicate enrichment peaks of ChIP-Seq signal for MECOM. **j** Location of inferred MECOM-to-NR2F2 binding site (in the blue box). The enlarged figure shows the location of MECOM binding signal, motif, and the four CRISPR gRNA locations. **k, l** Relative interaction frequency measured by 3 C assays. **m, n** Expression level of *NR2F2* and *NR2F2-AS1* in H1-hESC and HUVEC. **o, p** Relative gene expression levels of *NR2F2*. Data are presented as mean values $\pm$ SD; $n = 3$ (**j–l** and **o, p**) and $n = 2$ (**m, n**) biologically independent samples. *P* values are determined by a modified two-tailed Fisher Exact test implemented in DAVID v6.8 (**b, f**), two-tailed Fisher Exact test (**c, e**), two-tailed Poisson test (**h**), and two-tailed Negative Binomial test implemented in edgeR v3.14.0 (**m, n**), and two-tailed Student's *t* test (**k, l, o, p**). Source data are provided as a Source Data file.

several EC-related pathways such as EC differentiation (GO:0045446), EC fate commitment (GO:0060839), EC proliferation (GO:0001935), EC migration (GO:0043542), and endothelial development (GO:0003158). These MECOM target sites around *NR2F2* overlap with a strong chromatin loop in HUVECs. Further, this loop is lost in H1-hESC (Fig. 5i top, marked by black circles). Compared to ESCs, the *NR2F2* locus in HUVECs is marked with strong signals of activating histone modifications including H3K4me3, H3K27ac, and H3K4me1, but was devoid of the repressive mark H3K27me3 (Fig. 5i bottom). By analyzing MECOM Chip-seq data and binding motif, we located the MECOM binding site in the enhancer that is connected to the *NR2F2* promoter by the chromatin loop (Fig. 5i bottom, Fig. 5j bottom). Using CRISPR-cas9 system, we deleted this binding site in the enhancer using two pairs of gRNA sequences around this region (Fig. 5j top panel). We next performed chromosome conformation capture (3 C) assays based on 5 pairs of primers (Fig. 5j bottom panel, 3 C primer design was described in METHODS). We observed that the interaction frequency between the enhancer and promoter of *NR2F2* was significantly reduced by the deletion of the MECOM binding site (Fig. 5k). We further knocked down *MECOM* in HUVEC and then performed 3 C assays. The data showed that the interaction frequency between the enhancer and promoter of *NR2F2* was also significantly reduced by the knockdown of *MECOM* (Fig. 5l). Consistent with the increased expression level of *NR2F2* in HUVECs compared to ESCs (Fig. 5m, n), the RNA expression level of *NR2F2* was also significantly downregulated after we deleted the MECOM binding site or knockdown MECOM expression (Fig. 5o, p). These results verified that MECOM expression and the MECOM target site are required to form the chromatin loop and activate the transcription of *NR2F2*. We noted that 3 C experiment might have its limitation, as primer annealing efficiency can potentially bias the contact frequency. Therefore, we used 5 pairs of primers to mitigate this bias. Other comprehensive and high resolution methods such as 4 C or Hi-C might be helpful to further validate the observation in future. Taking together, both linear epigenetic landscapes analysis and 3D chromatin conformation experiments indicated that MECOM binds active enhancers to regulate target genes in ECs.

## VEGF signaling pathway is a key target of MECOM
VEGF signaling pathway, the principal pathway modulating angiogenesis[41], is a key determinant of EC differentiation and functions. EC differentiation from pluripotent stem cells[42] and transdifferentiation from fibroblasts[5] both could be induced in vitro by treatment with the VEGF. We found that the VEGF signaling pathway is enriched in the downregulated genes after *MECOM* depletion (Fig. 5b). Manual inspection revealed that 14 out of 28 nodes in the VEGF signaling pathway (KEGG: hsa04370) are significantly downregulated in *MECOM* KO HUVEC pool (Fig. 6a), suggesting the key role of *MECOM* in regulating this pathway. Importantly, we found the *MECOM* depletion downregulated *VEGFR2*, the hub gene of VEGF signaling (Fig. 6b). VEGFR2 is the prominent receptor of VEGF that mediates cellular

processes involved in angiogenesis[42,43]. Furthermore, several downstream pathways regulated by VEGF signaling are also significantly enriched with genes that are downregulated by *MECOM* depletion, e.g., proliferation, adhesion, and migration pathways (Fig. 6a). Manual inspection found that *MECOM* depletion significantly downregulated the hub genes in these pathways, e.g., the *FGF2* of Endothelial Migration pathway[44] (Fig. 6c), the *PDGFB* of Endothelial Proliferation pathway[45] (Fig. 6d) and the *ITGAV*[46] (Fig. 6e) of the Focal Adhesion pathway. Further, on the basis of RNA expression data from 11,688 healthy human samples across 53 tissue types in the GTEx database[47], our unbiased analysis revealed a global expression correlation between *MECOM* and *VEGFR2* (Spearman correlation coefficient 0.63) (Fig. 6f).

We further found that the genes associated with MECOM binding motifs were significantly enriched in the VEGF signaling pathway (Fig. 5f). Particularly, an enrichment peak of MECOM ChIP-Seq signal in SKOV3 cell suggested a MECOM binding site upstream of *VEGFR2*. This binding site overlaps with strong signal of activating histone modifications including the H3K27ac and H3K4me1 in HUVECs (Fig. 6g). In contrast, both signals of activating histone modifications were lost in ESCs (Fig. 6g). Further, the binding site is located in a TAD domain that contains only one gene, i.e., the *VEGFR2* (Fig. 6h), suggesting the *VEGFR2* is likely to be the direct target gene of this binding site. To further confirm the role of this binding site in the expression regulation of *VEGFR2*, we utilized CRISPR-Cas9 to knockout the binding site (Fig. S9a). PCR experiments verified the successful deletion of the MECOM binding site (Fig. S9b) by two CRISPR guide RNAs (gRNAs) flanking that region (Fig. S9a). We found that the deletion of the MECOM-to-VEGFR2 binding site significantly reduced *VEGFR2* mRNA (Fig. S9c) and protein expression (Fig. 6i). We further found that overexpression of MECOM could rescue the VEGFR2 down-regulation caused by MECOM KD (MECOM shRNA targeting MECOM 3′ UTR) but could not rescue the VEGFR2 down-regulation caused by the deletion of MECOM-to-VEGFR2 binding site (Fig. 6i), confirming that the MECOM binding site is required to activate the transcription of VEGFR2.

A previous study reported functional cooperation between MECOM and AP-1 in Hela cell[40]. Accordingly, we next determined if AP-1 plays a role in the regulation of *VEGFR2* by MECOM in HUVECs. After knocking down *AP-1* in HUVECs using shRNA (Fig. S9d), we found that binding of MECOM to the MECOM-to-VEGFR2 binding site is not affected (Fig. S9e) compared to wild type HUVECs. To confirm the specificity of MECOM-to-VEGFR2 binding site, we selected two non-MECOM binding sites as negative control sites, which are either 1.5 kb downstream (negative control-1) or 4.5 kb upstream of the MECOM-to-VEGFR2 binding site (Fig. S9a). These negative control sites both lack enrichment of MECOM ChIP-Seq signal and enhancer markers (H3K27ac and H3K4me1). We found both negative control sites showed no MECOM binding signal in MECOM ChIP-PCR experiment (Fig. S9f, g).

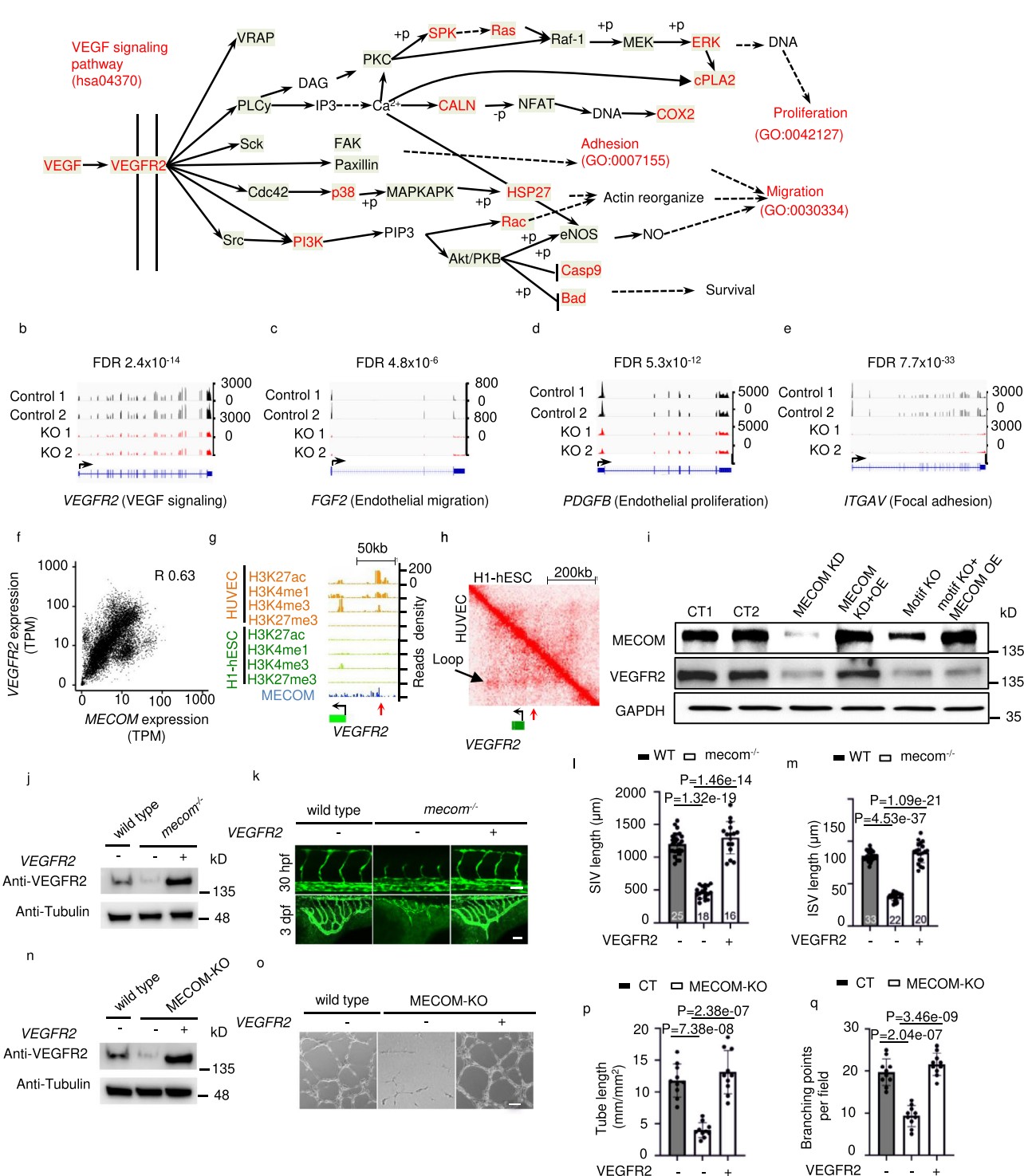

## VEGFR2 rescued the impaired angiogenesis in *mecom*-deficient zebrafish and HUVECs

To verify that MECOM regulates EC function by modulating VEGFR2 expression, we overexpressed *VEGFR2* by transfection of modified messenger RNA (mmRNA) after *mecom* KO in zebrafish embryos (Fig. 6j). The overexpression of VEGFR2 restored the impaired angiogenesis (Fig. 6k). Specifically, reduced SIV length (Fig. 6l) and ISV length (Fig. 6m) in *mecom* KO embryos of zebrafish were effectively restored. Similarly, VEGFR2 overexpression rescued the impaired tube formation in *MECOM*-depleted HUVECs (Fig. 6n, o). Specifically, both the decreased tube length (Fig. 6p) and branching points (Fig. 6q) are

restored to normal levels by VEGFR2 overexpression. These results indicate that overexpression of VEGFR2 reversed the defective angiogenesis caused by MECOM deficiency both in vitro and in vivo, verifying that VEGFR2 acts downstream of MECOM to regulate angiogenesis.

## Discussion

Prior reverse genetic studies to identify regulators of EC lineages generally require labor-intensive work associated with genetic screens. With the large set of epigenomic and genetic data available, we applied a different approach by developing a data-driven framework to

**Fig. 6 | VEGF signaling pathway is a key target of MECOM. a** The schematic representation of VEGF signaling pathway defined in KEGG. Pathway members which significantly downregulated after MECOM KO in HUVEC were marked in red color. **b–e** Signal density of RNA-Seq at the locus of *VEGFR2* (**b**), *FGF2* (**c**), *PDGFB* (**d**), and *ITGAV* (**e**) in both Control and MECOM KO HUVEC pool. **f** Gene expression correlation between *MECOM* and *VEGFR2*. **g** ChIP-Seq read density for chromatin marks and the MECOM protein at *VEGFR2* locus in HUVEC. Red arrow indicates the location of MECOM binding peak. **h** Two-dimensional heat map of chromosomal contact frequency at *VEGFR2* locus in HUVEC and H1-hESC. Red arrow indicates the location of MECOM binding peak as in (**g**). **i** Western blot analysis of protein levels of MECOM, VEGFR2, and GAPDH in HUVECs. **j** Western blot analysis of protein levels of Vegfr2 and Tubulin in wild type and *mecom* knockout mutants (*mecom⁻/⁻*) zebrafish injected without (−) or with (+) modified mRNA (mmRNA) of *VEGFR2*. **k** ISV angiogenesis (top) and SIV angiogenesis (bottom) in wild type, *mecom⁻/⁻* zebrafish injected without (−) or with (+) *VEGFR2* mmRNA. (**l-m**) Quantification of SIV length (**l**) and ISV length (**m**). **n** Western blot analysis of protein levels of Vegfr2 and Tubulin in control, *MECOM* KO HUVECs treated without (−) or with (+) *VEGFR2* mmRNA. **o** Tube formation of control or *MECOM* KO HUVECs pool transfected without (−) or with (+) *VEGFR2* mmRNA. **p, q** Quantification of tube length (**p**) and branching points per field (**q**). Scale bars, 100 μm. All values are indicated as mean ± SD (**l, m, p, q**); numbers of animals analyzed are indicated in the bars (**k, l, m**). *n* = 3 biologically independent samples (**i, j, n, p, q**). *P* values are determined by two-tailed Negative Binomial test implemented in edgeR v3.14 (**b–e**) and two-tailed Student's *t* test (**l, m, p, q**), Source data are provided as a Source Data file.

systematically annotate epigenetic features pertinent to cell identity genes, and to prioritize candidate cell identity genes. Our results indicated that the broad enrichment of activating epigenetic marks is better than the intensity of gene expression to serve as a molecular signature of EC identity genes. This observation may be because transcriptional regulation at each gene can be depicted at a high resolution by epigenetic marks, whereas the intensity of expression is the final output of a set of complex regulatory processes that may or may not be related to cell identity. As a result, the epigenetic landscape might more faithfully denote a cell identity gene. The major difference between cell identity genes and other expressed genes is that cell identity genes employ unique epigenetic mechanisms to regulate their transcription. By contrast, housekeeping genes are highly expressed, but often show sharp rather than broad enrichment of activating epigenetics marks[13]. Besides, we would like to emphasize that the association of broad activating epigenetic marks and cell identity gene is not exclusive to ECs. EC cell identity genes may include two broad gene groups, i.e., genes required for the establishment of EC identity and genes required for the maintenance of EC functions. Specifically, pattern of epigenetic marks can be used to predict EC cell identity genes but is not powerful enough to distinguish whether a gene is required for EC lineage determination or EC function maintenance. Accordingly, we further performed comprehensive experimental analysis to verify the function of the top candidate cell identity gene, MECOM, in both EC lineage determination and maintenance of EC functions.

As an important transcription factor, MECOM plays an essential role in regulating gene expression during early development and hematopoiesis[48]. *Mecom* (also called EVI1 in many literatures) mutant mice showed severe heart malformations, including failing heart, delayed chamber development, and poor general circulation[49]. Mice carrying a hypomorphic *Mecom* allele also exhibited severe congenital heart defects, including common arterial trunk, double outlet right ventricle, and aortic arch formation impairments[50]. These types of congenital heart defects represent the major cause of the perinatal lethality seen in *Mecom* mutant pups. *Mecom* is also a critical regulator for hematopoietic stem cells (HSC) and transformed leukemic cells[48]. *Mecom* mutant exhibited defective HSC activity in *Mecom*-deficient embryos. Selectively disrupting *Mecom* function in Tie2+ endothelial and hematopoietic stem/progenitor cells caused a defect that mimics EC deficiency[48]. Although the defects of the heart and vessels were not the primary focus of these studies, the researchers did reveal that many *Mecom* -/- embryos exhibited defective large vessel development after E13.5[48].

Although *Mecom* mutant mice showed severely impaired heart and vessel development, the underlying mechanism is largely unknown. We recently developed a machine learning framework CEFCIG, which also identified MECOM as an EC identity regulator required for endothelial differentiation[18]. More recently, another group also observed high expression of *MECOM* in arterial EC and demonstrated that *MECOM* KD in hESC-derived EC upregulates non-arterial EC markers[19]. However, the underlying molecular mechanism of MECOM in regulating EC identity is largely unknown. In the current

work, we combined epigenetic landscape analysis and experimental verifications, and uncovered that MECOM is a top regulator in the network of cell identity genes for ECs. *MECOM* displays typical epigenetic signatures for an EC identity gene. It possesses all three well-known cell identity gene signatures, including bivalent promoters in ESCs as well as broad H3K4me3 and super-enhancers in ECs. In addition, *MECOM* shows other epigenetic patterns highly relevant to cell identity genes, including broad enrichment of other activating histone modifications, broad open chromatin conformation, progressively increased expression level during EC differentiation from ESCs, and the exclusive expression of *MECOM* in the cell cluster of bona fide iPSC-ECs. The importance of MECOM as an EC identity gene was confirmed by multiple endothelial assays. Disruption of MECOM impaired the differentiation of human iPSCs to ECs, impaired the function of human ECs, and abrogated angiogenesis in zebrafish. Our RNA-Seq, ChIP-Seq, Hi-C, and DNase-Seq analyses at the genome-wide scale revealed that MECOM bound enhancers that formed chromatin loops with genes important for EC differentiation and functions. Therefore, we annotated the role of MECOM on the basis of robust prioritization and extensive experimental and genome-scale validation. These data together highlight MECOM as a lineage regulator of EC.

Considering prior findings that HSPCs and ECs are developed from common precursors[1], together with the known role of MECOM in HSPC[34] and our new data, it is likely that MECOM regulates both angiogenesis and hematopoiesis in the circulatory system. MECOM was identified as an oncogene in myeloid leukemia[51], and was critical for the endothelial to hematopoietic transition[34]. A study reported that *Mecom* homozygous mutant disrupted the development of paraxial mesenchyme in E10.5 mouse embryos, and suggested that *Mecom* has important roles in vascularization at midgestation[49]. In contrast, MECOM impairs the differentiation of granulocytic and erythroid lineages from hematopoietic precursors[39,51]. Our data demonstrated that *MECOM* displayed epigenetic signatures of cell identity genes in arterial ECs and venous ECs, whereas these signatures are attenuated in lymphatic ECs. Taken together, literature and our data suggest that MECOM promotes early or middle stage differentiation of mesoderm along the endothelial and hematopoietic lineages in the circulatory system, but is less likely to be required for the terminal differentiation along these lineages.

To conclude, we have developed a landscape analysis of epigenetic patterns at regulators for EC identity. The analysis revealed distinct epigenetic patterns underlying EC lineage specification as well as the specification of EC subtypes. This analytical approach allowed us to reveal MECOM as a leading candidate for EC lineage regulator. Further investigations indicated that MECOM bound enhancers to regulate EC gene expression, e.g., the transcription of *VEGFR2*. Finally, we demonstrated that MECOM was required for EC differentiation, EC functions, and vessel formation.

## Methods

### Ethical regulation

Animal care and use conditions were followed in accordance with institutional and National Institutes of Health protocols and guidelines.

All studies were approved by Houston Methodist Institution Animal Care and Use Committee and under appropriate project protocols.

## ChIP-seq and DNase-seq analyses

We downloaded the Human reference genome sequence version hg19 and UCSC Known Genes from the UCSC Genome Browser website[52]. We used Bowtie v1.1.0 to map sequencing reads to the human reference genome, requiring single best match for each read across the genome. We used the Dregion function in DANPOS v2.2.3 to calculate reads density from the mapped reads as the number of reads covering each base pair of the genome and to define enriched peaks with cutoff being Poisson test $P$ value 1e−5 for seed peak calling and 1e−3 for peak extending. For each dataset, we normalized the total number of mapped reads to 25 million and extended each read at the 3′ end to be 200 bp long. The function Dregion in DANPOS v2.2.3 also accounts for variations between biological replicates of the same ChIP experiment. We set the bin size to 10 bp and the smooth width to 0 bp to not use any smoothing step in the calculation of reads density. Input effect was subtracted from the ChIP-Seq data by DANPOS v2.2.3. For reference gene set, we downloaded the KnownGene provided at the Table Browser page of UCSC Genome Browser (http://genome.ucsc.edu/cgi-bin/hgTables). We used the function Profile in DANPOS v2.2.3 to plot average ChIP-Seq reads density around all transcription start sites (TSS) or gene body and generate data matrices for heat maps of ChIP-Seq reads density around each TSS or gene body, and used the tool MultiExperimet Viewer (MEV v10.2) to plot the heat maps. To map peaks to individual genes, we used the function Selector in DANPOS v2.2.3 to retrieve peaks that are on gene body or within 10 kb distance to either TSS or transcription termination site (TTS) in intergenic region. The sum of widths of all peaks belonging to a gene is calculated as gene level peak width.

## RNA-Seq analysis

HUVEC (Lonza, #CC-2517) under wild type control and *MECOM* depleted condition were subjected to RNA-Seq, with two replicates for each condition. We used TopHat v2.0.12 to map the raw reads in FASTQ format to the hg19 human reference genome with the following parameters settings: --mate-std-dev 200 -p 8 -r 200. The mapped reads for each sample were saved in a BAM format file and we used UCSC Known Genes as reference genes. The BAM file and reference genes were subjected to the Cuffdiff function in Cufflink suite v2.2.1 to calculate read counts and gene expression (fragments per kilobases per million, or FPKM). To identify differentially expressed genes based on read counts between different RNA-Seq samples, we used the normalizeQuantiles, estimateCommonDisp and estimateTagwiseDisp functions in the R package edgeR v3.14.0 to normalize the read counts, estimate common dispersion and estimate moderated tag-wise dispersion, respectively. The edgeR then defined differential genes based on a negative binomial test. In the final list of differential genes, we required each differential gene to have a differential FDR adjusted $p$ value ($q$ value) smaller than 0.05 and FPKM value larger than 1 in at least one sample. To calculate RNA-Seq read density at each base pair in the genome, we used the genomecov function in Bedtools v2.17.0 to convert the mapped reads in a BAM file to read density values saved in a BedGraph format file. After normalizing the total density values in each sample to 10 billion using a custom python script, we used the tool bedGraphToBigWig v4 to convert the normalized BedGraph file to a bigWig format file. We then subject the bigWig file to the Integrative Genomics Viewer (IGV) v2.3.67 to plot the read density tracks.

## Hi-C analysis

To examine the 3D chromosomal organization, we downloaded the compressed contact matrix (hic file) for HUVECs[53] and H1-hESC (4DN Data Portal, https://data.4dnucleome.org/). The raw Hi-C sequencing data was processed using Juicer[54] and aligned against the hg19

reference genome. Both contact matrices used for downstream analysis were KR-normalized with Juicer. We visualized the contact map in hic format using Juicebox[55]. Contact maps of HUVECs and H1-hESC are browsed side-by-side in 2D heat map and compared with each other to highlight regions showing differential chromatin loop structures. To compare the 2D contact heat maps to other 1D epigenetics tracks, including histone modifications, CTCF and RNA-Seq, we collected the bigwig track of these epigenetic marks for both HUVECs and H1-hESC from ENCODE project[9] and loaded them in Juicebox.

We annotated chromatin loops across both contact maps using HiCCUPS[54]. With default parameters of HiCCUPS, chromatin loops were called at 5 kb, 10 kb, and 25 kb resolutions and merged as described previously[53].

To examine the association between MECOM target sites and chromatin loop regions, an overlapping MECOM target site is defined as a site that has at least 1 bp overlap with either region of a pair of contacting regions brought together by a chromatin loop.

## Quantitative analysis of chromosome conformation capture assays (3C-qPCR) and probe design

3C-qPCR was performed following Dr. Forne's protocol[56]. The 5 pairs of primers used to quantify the interaction frequency were designed according to the following procedures:

We downloaded BAM files of Hi-C data with two replicates generated in HUVEC cell line from the 4DN data portal. The accession numbers of the two files are 4DNFIA5OQ5P7 and 4DNFIGKQDTOR. The BAM file of the Hi-C data contains the paired-end read sequences as well as the genomic locations that they were mapped to. In order to obtain the reads of our interest from the BAM files, we first located the genomic coordinates of the enhancer and promoter of *NR2F2*. Next, we extracted the paired-end reads with one end mapped to the enhancer region and the other end mapped to the promoter region. The collected reads from the Hi-C BAM files of the two replicates were merged. Since Hi-C sequenced both ends of the DNA from ligated enhancers and promoters, it happens that one end was sequenced only for enhancer or promoter, while the other end was sequenced across the ligation site which usually results in a partial mapping to the reference genome for one of the read ends but not the other. Particularly, the DNA sequences of reads from which one end was perfectly mapped while the other was partially mapped were used as the candidate's template to design the 3 C primers.

## scRNA-seq data analysis

For scRNA-seq on differentiating iPSCs at days 8 and 12 of EC differentiation (GSE116555), we downloaded the *rda* file that stores Seurat format of the processed data generated by the original authors. We carried out analyses of processed scRNA-seq data in R version 3.5.1 with Seurat v2.0[57,58]. To examine transcriptome heterogeneity and find distinct cell clusters, we performed principal component analysis (PCA) to reduce data dimensionality. For both samples, we selected the top 20 significant principal components using a permutation-based test and heuristic methods implemented in Seurat. The selected PCA loadings are used as input for graph-based cell clustering[59] and as input for t-distributed stochastic neighbor embedding (t-SNE)[60] for reduction to two dimensions for visualization purposes. To track the cell dynamics during EC differentiation from iPSCs, we performed a canonical correlation analysis to identify common sources of variation between the dataset for 'Day 8' and 'Day 12', followed by dimension reduction, cell clustering, and visualizations.

## Function enrichment analysis

We used DAVID v6.8 (https://david.ncifcrf.gov)[61] for Gene Ontology pathway analysis. Each terms of Gene Ontology Biological process with $P$ value smaller than 0.05 is defined as significantly enriched.

## Assign peak and motif to target genes

MECOM can bind to regulatory elements of target genes, including both proximal promoter regions and distal enhancer regions. Enhancer regions could potentially be far away from their target genes and it is biologically hard to justify an upper limit, we decided to provide a more global review of MECOM enrichment profiles without choosing arbitrary cutoff for the upper limit of distance between enhancer and its target genes. This analysis was performed by the Cumulative Analysis of Genomic Region Enrichment (CAGRE) tool developed by us https://github.com/jielv/CAGRE[62].

## Motif analysis

We used HOMER v4.10 to infer de novo MECOM biding motif from MECOM ChIP-Seq data and detect the inferred MECOM binding motif instances around gene promoters. To account for the potential difference of MECOM binding landscape between Cancer cell (SKOV3) and HUVEC cells, we selected MECOM binding peaks locating at HUVEC open chromatin regions (by overlapping with DNase Peaks from HUVECs) as "HUVEC-active MECOM peaks", and only use this subset of peaks for motif finding analysis. We used *findMotifsGenome.pl* program of HOMER to infer de novo MECOM binding motif using 400 bp regions around the center of MECOM binding peaks. For MECOM motif instance detection, we scanned the MECOM motif position weight matrices (PWD) using *annotatePeaks.pl* program of HOMER to find instances of motifs near MECOM ChIP-Seq peaks. Motif density profile across MECOM peaks with positive hits is generated using the "-hist" parameters of *annotatePeaks.pl* program and home-made python script.

## Gene regulatory network (GRN) analysis

To evaluate the effectiveness of our method in identifying EC identity genes, we quantitatively compared the performance our method to those of other ones based on gene expression data using Gene Regulatory Network (GRN) defined in the database of the software CellNet. Each edge of GRN represents the existence of gene regulatory relationship between the two genes it connects. And GRN illustrates the regulatory mechanism of a given set of genes and provides insights on the synthesizing effects of different genes in cells. In detail, we quantified the number of GRN edges connecting the top 400 genes from our histone modification-based method as well as from other cell identity gene prediction methods, including the ones based on absolute expression level, expression level differentiation, and simulation of random gene selection. The group of genes showing higher number of GNR edges are more likely to be functionally relevant to each other, implying a higher probability to form a gene regulatory network underlying the regulation of EC cell identity.

To quantify the role of individual candidate gene in EC Gene Regulation Network (GRN), we reconstructed a GRN using the list of potential EC identity genes defined by each method. We measured the number of unique paths going through a given gene and its neighbor genes, a centrality indicator of the examined gene, to quantify the participation of a given gene in the inferred EC GRN. Accordingly, a gene with high centrality displays a strategic position in the EC cell identity GRN, suggesting it may play a master role in regulating EC biological functions.

## Protocol for iPSC differentiation to endothelial cells

For EC differentiation, human iPSCs (gift from Dr. John Cooke's lab) were plated on growth-factor-reduced Matrigel in mTeSR1. After 24 h, the medium was replaced with mesoderm induction medium (DMEM/F12 medium supplemented with 30 ng/ml BMP4 and 1.5 μM CHIR) for 2 days. After that, an endothelial specification medium (EGM2 medium supplemented with 50 ng/ml VEGF and 10 μM SB431542) was used for a total of 4 days and replaced every 2 days. On day 7 of differentiation,

ECs were dissociated for FACS analysis or sorting for functional analysis.

## MECOM KO iPSC single colony generation

CRISPR Cas9/gRNA lentivirus particles targeting a CTCAAGTACATTAGATTCGC sequence of *MECOM* gene was generated using a method similar to previously reported[63]. CT lentivirus particles do not contain the gRNA sequence. WT iPSCs were infected with lentivirus particles for 4 h, and after 24 h, were selected with 1ug/ml puromycin for 24 h. iPSCs were then seeded on 96-well plates to get single cell colonies. Colonies were screened with PCR and TA cloning sequencing. Heterozygous *MECOM* mutant iPSC colonies were infected once again with CRISPR Cas9/gRNA lentivirus particles targeting *MECOM* and selected for *MECOM* KO iPSC single colonies.

## Flow cytometric analysis

After EC differentiation, cells were trypsinized, centrifuged at $200 \times g$ for 5 min, resuspended in FACSB-10 (FACS buffer with 10% FBS), incubated with CD144 monoclonal antibody (Alexa fluor 488, Catalog # 53-1449-42, 1:100 dilution in FACSB-10) and CD31 monoclonal antibody (APC, Catalog # 17-0319-42, 1:100 dilution in FACSB-10) for 30 min on ice. Fluorescence was determined using a flow cytometer (LSR II, Becton Dickinson, San Jose, CA, USA), and the data was analyzed using FlowJo software.

## Stable HUVEC KO cell pool generation

HUVEC cell culture and lentivirus infection were performed as previously described[64] with minor modifications. Briefly, HUVECs were cultured in Endothelial Growth Medium (EGM, Cat. No. CAP02, Angio-Proteomie) and HUVECs at passage 4 were grown to 60% confluence for infection with lentiviral particles. The plasmid lentiCRISPR-V2 (Addgene, #52961) with or without human *MECOM* gRNA target site sequences TGTGGGTGAAACAAGAATCC (g1) or CTCAAGTACATTAGATTCGC (g2) were co-transfected with psPAX2 (Addgene, #12260) and pMD2.0 (Addgene, #12259) into 293-T cells (ATCC, #CRL-3216) for 4 days, lentiviral particles were collected and used to infect HUVECs. Stable control or *MECOM*-KO cell pool were obtained by puromycin (1 μg/ml) selection for consecutive 14 days after viral infection. *MECOM* KO was confirmed by protein expression analysis.

## HUVEC proliferation/viability, tube formation, scratch migration, and NO measurement

To assess cell proliferation/viability, HUVEC after viral infection were seeded in a 96-well plate at a density of 3000 cells per well in Endothelial Growth Medium. Cells were allowed to attach for 24 h and then cell proliferation/viability was monitored utilizing CellTiter-Glo® Luminescent Cell Viability Assay (Promega, Madison, WI, USA) for 24–72 h. Results were read at 24, 48, and 72 h on the on Synergy 2 Multi-Mode Reader. HUVEC tube formation and scratch migration were performed as previously described[64]. In brief, 48 well plates were coated with 100 μl matrigel (Corning, Cat. No. CB-40234A) and incubated at 37 °C for 30 min. $5 \times 10^4$ control and *MECOM*-KO HUVEC pool in 125 μl EGM medium were seeded in each well, respectively. After 12 h, images were captured using the Leica epi-fluorescence microscope. Migration assay was performed by scratching the confluent HUVEC monolayer with a p200 pipette tip. Wound closure was monitored using digital photography at 0 h, 6 h, and 12 h later and measured using the NIH Image J program. The number of migrating cells was quantified by counting cells that crossed into the scratched area from their reference points. Griess assay was used for NO measurement based on the manufacturer's manual (cat. No. G-7921).

## Fluorescent LDL uptake

Control and *MECOM*-KO HUVEC pool were seeded on 48 well plates. Alex Fluor 594 AcLDL (Thermo Fisher Scientific, Cat. No. L35353, 1:100

dilution in EC medium) was added to the culture medium for the final 4 h of the incubation time. HUVECs were washed, trypsinized, and centrifuged at $200 \times g$ for 5 min, then resuspended in FACSB-10. Fluorescence was determined using a flow cytometer (LSR II, Becton Dickinson, San Jose, CA, USA), and the data were analyzed using FlowJo software.

## Generation of *mecom* mutants zebrafish using CRISPR/Cas9

Zebrafish *Tg(fli1:EGFP)^y1* fish lines were bred and maintained at 28.5 °C on a 14 h light/10 h dark cycle[65]. The CRISPR/Cas9-mediated gene knockout technology in zebrafish was performed as previously described[64,66]. Briefly, the Cas9 capped mRNA was synthesized using mMESSAGE mMACHINE SP6 kit (Thermo Fisher Scientific, Cat. No. AM1340); guide-RNA (gRNA) was synthesized using HiScribe T7 Quick High Yield RNA Synthesis kit (NEB, Cat. No. E2050S); the target sequences for *mecom* were 5′- GGTGGGTGGATGGTGAGATC-3′. The gRNA (50 ng/ul) and Cas9 mRNA (150 ng/ul) was mixed and injected into one-cell stage *Tg(fli1:EGFP)^y1* embryos.

For the genotyping of *mecom* mutants, genomic DNA was extracted from individual zebrafish embryo and a 457 bp genomic region flanking the target site was PCR amplified. The primers used for genotyping were mecom-F (5′- GCCTCAGTGGGTGGCATGAG-3′) and mecom-R (5′-TGTGATTATCCTCATGCTTGACCG-3′). Purified PCR products were denatured and re-annealed, and then digested with T7 Endonuclease I (NEB, Cat. No. M0302S). PCR products were sequenced to confirm the frameshift of *mecom* gene.

## Genotyping of germline mutants

To assess germline inheritance of genome modification generated in somatic mutant fish, P0 founders from *mecom* sgRNA injection experiment were outcrossed to WT zebrafish and genomic DNA was extracted from F1 larvae or tailfin clips of adult F1 fish. DNA from individual fish was used as a template for subsequent PCR using primers spanning the target site of *mecom* sgRNA (described above). PCR amplicons encompassing the target region were analyzed by Sanger sequencing to detect sequence variants. F2 larvae were obtained from F1 inbreeding and used for all the experiments in this study. The same primers were used for PCR amplification of genomic DNA in order to discriminate WT, heterozygous and homozygous carriers of the *mecom* allele among the offspring of subsequent inbreeding experiments.

## Whole-mount alkaline phosphatase staining

Endogenous alkaline phosphatase staining was performed according to the previous reports[64,67]. In brief, zebrafish larvae at 3 days post-fertilization were fixed in 4% paraformaldehyde (PFA) at room temperature for 2 h. The embryos were dehydrated and suspended in 100% methanol overnight. Then the embryos were rehydrated and finally suspended in PBST (phosphate buffered saline with 0.1% Tween-20, pH 5.5). After equilibrated with alkaline phosphatase buffer, the embryos were stained using the NBT/BCIP solution at room temperature for 20 min. Embryos were mounted in glycerol for imaging analysis.

## Confocal imaging and data analysis

Anaesthetized embryos at indicated stage were mounted in 1.0% low-melt agarose for imaging. Images were acquired by using an Olympus FluoView FV1000 laser scanning confocal microscope. Z-stacks were acquired with a 3 μM step, and images were 3D rendered. Neurolucida software (MBF Bioscience) was used to analyze the length of inter-segmental vessels (ISVs) and subintestinal vein (SIV).

## CRISPR single-guide RNA (sgRNA) preparation

Two open-access software programs, Cas-Designer (http://www.rgenome.net/cas-designer/) and CCTop (https://crispr.cos.uni-heidelberg.de) were used to design guideRNAs (gRNA) targeting MECOM binding sites. For MECOM-to-VEGFR2 binding site, we targeted a 70–80 bp *VEGFR2* upstream enhancer region that include a 8 bp MECOM binding motif. The two gRNA sequences are "ATTCTCATTAAAATCCTGTG" and "AGCTGGTGACTCACAA-ACCA" respectively. Then we used DNA primer *MECOM*-PCR-3F: CATCAGGCCCTGTGCTAAG and *MECOM*-PCR-3R: CTCGGATGCCTTCTCTTCCT to amply this region. For MECOM-to-NR2F2 binding sites, we targeted a 200 bp *NR2F2* upstream enhancer that includes 2 MECOM binding motif. The gRNA sequences we used were: g1: ATGTAACTTGCGCTGCATTGAGG; g2: TGTTGTATAGCCTCAATAACAGG; g3: AGTGGTTGTTACTGCTGTGTGGG; g4: ACACAGCAGTAACAACCACTCGG. Target DNA oligos were purchased from IDT (Integrated DNA Technologies) and cloned into the lentiCRISPR v2 plasmid (Addgene plasmid# 52961) via BsmBI restriction enzyme sites upstream of the scaffold sequence of the U6-driven gRNA cassette. All plasmids were sequenced to confirm successful ligation.

## RNA isolation and qPCR

Total RNA was isolated from adult zebrafish or cultured cells using the Quick-RNA Mini-prep kit (Direct-zol, R2052, ZYMO Research, Irvine, CA, USA) and cDNA was obtained using amfiRivert cDNA Synthesis Platinum Master Mix (R5600-100,5 GenDEPOT, Barker, TX, USA). Each cDNA sample was amplified using Power SYBR Green PCR Master Mix (BIO-RAD, Hercules, CA, USA) on the QuantStudio 6 Flex Real-time PCR System (403115082, GE Healthcare). Briefly, the reaction conditions consisted of 2 μl of cDNA and 0.2 μM primers in a 10 μl final volume of super mix. Each cycle consisted of denaturation at 95 °C for 15 s, annealing at 58.5 °C for 5 s, and extension at 72 °C for 10 s, respectively. *GAPDH* was used as an endogenous control to normalize each sample. The primers against *KDR* are: primer #1 forward-AGGAATCCCTTTGCAAGGTT, reverse-CCCAAAGTGCTGGGTTTTTA; primer #2 forward-TGACTGCACAAACCAGCTTC, reverse-TGACACCACACACAGCTTCA. Those used in zebrafish samples were *mecom*-qPCR-F (5′- CCCTCTAATCCCATC-CACATC-3′) and *mecom*-qPCR-R (5′- CGCTCCATATTCTCGCTTTC-3′), *ef1a*-qPCR-F (5′-ACCGGCCATCTGATCTACAA-3′) and *ef1a*-qPCR-R (5′- CAATGGTGATACCACGCTCA-3′).

## Western blot assay

To extract protein, cells were treated in cold RIPA buffer containing protease inhibitor. To denature proteins, lysates were added to 4× loading buffer and heated to 95 °C for 10 min. Total cell lysate (30–50 μg) were loaded onto SDS PAGE gels, and then transferred to PVDF membranes. Blots were incubated with primary antibodies overnight at 4 °C, followed by detection with secondary antibody. The antibodies used were: MECOM (Cell Signaling Technology, #C50E12, 1:500 dilution in PBST), beta-Actin (Cell Signaling Technology, #4967, 1:1000 dilution in PBST), KDR/VEGFR2 (Santa Cruz, #sc-6251, 1:500 dilution in PBST), GAPDH (Santa Cruz, #sc-32233, 1:1000 dilution in PBST), alpha Tubulin (Santa Cruz, #sc-5286, 1:1000 dilution in PBST), AP1(Signaling Technology, #9165, 1:1000 dilution in PBST), goat anti-rabbit HRP-conjugated antibody (Jackson Labs, #111-035-144, 1:5000 dilution in PBST).

## VEGFR2 rescue assay

*VEGFR2* and mCherry modified mRNA (mmRNA) were synthesized by Houston Methodist RNA Core facility. 50 pg *VEGFR2* mmRNA was injected into each one-cell stage embryo. The same amount of mCherry mmRNA was injected into wild type embryos as control. In control or *MECOM* KO HUVEC pool, $3 \times 10^6$ cells were electroporated with 6 μg VEGFR2 or mCherry mmRNA. Cells were collected and subjected to tube formation assay 48 h after electroporation.

## Alkaline phosphatase (AP) live staining

Alkaline phosphatase live staining kit (Invitrogen, #A14353) was used to measure the pluripotency of iPSC. Briefly, 1× working solution was used by diluting the 500X stock solution of the Live AP substrate in basal media. WT and *MECOM* KO iPSC (single clone) were incubated with the substrate for 20 mins, and washed twice with the basal media to remove excess substrate. Images were captured under fluorescent microscopy using a standard FIT-C filter.

## Statistical test

Two-tailed Wilcoxon test, Fisher Exact test, and Student's *t* Test are performed in R v4.0.2 using function *wilcox.text* with parameter *alternative = 'two.sided'*, *fisher.test* with parameter *alternative = 'two.sided'*, and *t.test* with parameter *alternative = 'two.sided'* respectively.

## Reporting summary

Further information on research design is available in the Nature Portfolio Reporting Summary linked to this article.

## Data availability

The RNA-Seq datasets for wild type HUVECs and HUVECs with *MECOM* depleted by CRISPR-Cas9 were generated in this project and deposited to the GEO database by the accession number GSE160647. Other genomic datasets were downloaded from public database. Database accession numbers for all datasets analyzed in this project were indicated in Supplementary Data 4. Source data are provided with this paper.

## Code availability

ChIP-seq and DNase-seq analyses were performed with Bowtie v1.1.0, DANPOS v2.2.3 and Multiple Experiment Viewer (MeV) v10.2. RNA-Seq analyses were performed with TopHat v2.0.12, Cufflink suite v2.2.1, edgeR v3.14.0, bedtools v2.25.0, bedGraphToBigWig v4, and Genomics Viewer (IGV) v2.3.67. Hi-C data analyses were processed using Juicebox v1.11.08 and juicer_tools v1.23.03. scRNA-seq data were processed with Seurat v2.0 installed on R version 3.5.1. Gene Ontology pathway analyses were performed with DAVID v6.8. Motif analyses were performed with HOMER v4.10. ChIP-Seq peaks and motifs of transcription factors (TFs) are assigned to target genes using Cumulative Analysis of Genomic Region Enrichment (CAGRE) v1.0 (https://github.com/jielv/CAGRE). Gene Regulatory Network (GRN) analysis is performed using custom scripts (https://github.com/jielv/GRN_analysis). Two-tailed Wilcoxon test, Fisher Exact test, and Student's *t* Test are performed using R v4.0.2.

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

## Acknowledgements

This work is supported by grants from NIH/NIGMS (R01GM125632 and R01GM138407 to K.C.) and NIH/NHLBI (R01HL133254 to J.C. and K.C.; R01HL148338 to J.C. and K.C.; 1R01HL132155 to L.F.; and R01HL155632 to L.Z.), and Additional Ventures (Single Ventricle Research Foundation to L.Z.), George and Angelina foundation to L.F., and Cancer Prevention Research Institute of Texas (CPRIT) grants RP150611 and RP170002. Q.C. is supported by U.S. Department of Defense (W81XWH-15-1-0639 and W81XWH-17-1-0357), American Cancer Society (TBE-128382) and NIH/NCI (R01CA208257).

## Author contributions

K.C. and L.Z. conceived and designed the project. J.L., X.G., and R.Z. interpreted the data and performed the data analysis. L.Z., K.C., L.F., and J.C. designed the experiments and interpreted the data. Q.G. and J.K. performed zebrafish experiments. S.M built MECOM KO iPSC single clone and performed EC differentiation experiments. M.C., Y.Z., S.Z., and Q.M. contributed to the cellular experiment. J.L., K.C., and L.Z. wrote the paper with input from L.F., Q.G., S.M., S.Z., and with comments from Q.C., J.C., B.X., G.W., D.Z., Y.L. Q. M., and X.W.

## Competing interests

The authors declare no competing interests.
