## [Peer Review File · Nature Communications]

Epigenetic Landscape Reveals MECOM as an Endothelial Lineage RegulatorREVIEWER COMMENTS

Reviewer #1 (Remarks to the Author):

Lv et al present a manuscript in which they identify MECOM as regulator of endothelial cell differentiation. This is done by leveraging a large number of ChIPseq datasets in HUVECs (human umbilical vein endothelial cells). From these difficult to grasp analyses MECOM is found as a key regulator. The authors go on to show using functional experiments that MECOM indeed is a functional regulator of endothelial differentiation. They generate knock-outs of MECOM in human iPSCs, which fail to differentiate to endothelial cells. In zebrafish knock-out of mecom results in impaired formation of vasculature. Finally, the authors show that MECOM regulates VEGFR2 through a distal regulatory element. Deletion of this element prevents the upregulation VEGFR2 in HUVEC. Overexpression of VEGFR2 in mecom knock-out zebrafish can rescue the impaired vasculature, suggesting that MECOM regulates endothelial differentiation through VEGFR2.

A novel regulatory role is presented for MECOM in endothelial differentiation. The functional experiments are very nice and presented in a clear manner. The rationalization for MECOM based on the ChIPseq data difficult to follow and not the most exciting part of the manuscript.

I have one major concern with the manuscript. The authors claim that MECOM target sites form loops to regulate target genes. The wording is already very suggestive, because this is all based on correlation. I suggest the authors delete the distal MECOM binding site at the NR2F2 locus (Figure 6l) and perform a 4C analysis to see whether loss of this binding site leads to a change in looping and gene expression.

Other points:

*The authors describe MECOM as a master regulator and it is clear that loss of this factor impairs the endothelial development, however, loss of MECOM can also be rescued by the overexpression of one factor. Can it be justified to call this a master regulator?

* In the MECOM KO iPSCs OCT4 and SOX2 remain expressed. Is pluripotency maintained? Can the authors do classical analyses to measure pluripotency (for instance alkaline phosphatase staining)? Related to this: it is strange that the OCT4 and SOX2 levels are normalized to the KO instead of the control.

*Figure 1h: p-values are not a measure for enrichment, a low p-value does not denote strong enrichment, but could just mean a large sample size. Please plot the enrichment scores instead of p-values.

*l. 166: the authors talk about hub genes and gene regulation network, but it is completely unclear where this comes from.

*l. 171: "our method", what is our method, this is not clearly defined.

*ref. 25: please cite the original research paper, and not only a preview paper.

*Figure 6c is unreadable

*Figure 7r: there is no 7r in the figure set.

Reviewer #2 (Remarks to the Author):

In this manuscript, Lv et al leverage available epigenomic and transcriptomic datasets to curate and identify novel regulators of endothelial identity. From this effort, the authors identify MECOM as an endothelial restricted transcription factor with several important functions in endothelial development and function. The authors show that loss of MECOM function results in impaired endothelial development in fish, defective endothelial cell differentiation from iPSCs, and disrupted endothelial cell functions in cultured endothelial cells. Additionally, the authors identify many endothelial genes and regulatory elements downstream of MECOM, including numerous components of the VEGF signaling pathway, and they infer the MECOM binding site from existing ChIP-seq datasets.

The identification of MECOM as a regulator of endothelial cell specification and differentiation would be an important advance and if confirmed to be true (see below) will be an important contribution to the field. Moreover, the approach of using existing datasets to identify cell specific regulators, particularly leveraging epigenomic datasets, is important and creative, and I believe would appeal to readers. Thus, in principle, I think this work could be suitable for publication in Nature Communications. However, there are several very significant issues, both technical and with regard to the existing published literature that affect the novelty and interpretation of the current work that absolutely should be fully addressed prior to any consideration for publication.

1) The authors primarily use data generated from HUVECs to identify MECOM as an upregulated transcription factor in endothelial cells and all comparisons are done mainly with respect to differential signatures of chromatin marks in ES cells. However, it is well known that MECOM is highly expressed in hematopoietic stem cells. It is also widely established that hematopoietic and endothelial cells share a common progenitor and are closely related cell types. This is not addressed in the manuscript.

1a) Moreover, the authors should compare MECOM expression in hematopoietic progenitors as well as chromatin signatures in these cells would help clarify enrichment levels on MECOM in endothelial cells in comparison to its expression level and signatures in known cell types.

1b) In Fig.3a-d, the authors analyze scRNA seq data from Day 8 and Day 12 of iPSC to EC differentiation and show that MECOM is expressed in Cluster 0, which they identify as endothelial cluster based on marker identity. However, most of the markers used to determine endothelial identity are also expressed in hematopoietic cells. The authors need to show that this cluster does not express other hematopoietic progenitor markers such as Runx1, CD41 or CD45 and is truly an endothelial cluster.

1c) If there are scRNAseq data available for earlier timepoints of this experiment (day 0 and day 4), it would be useful to show when MECOM expression begins and whether it precedes the expression of other endothelial markers.

2) Previous studies have shown that loss of MECOM in mice, embryonically as well as postnatally, primarily affects hematopoiesis with little or no effect on endothelial development. Additionally, conditional loss of MECOM in the endothelial lineage using Tie2-Cre results mainly in hematopoietic defects (not endothelial defects), suggesting that MECOM is not essential to maintain endothelial identity. The authors should discuss these published studies in greater detail and reconcile their data with these previously published papers. Given the lack of requirement for MECOM for endothelial cell development in mice, the authors may be overstating the role of MECOM as "a master regulator" of the endothelial lineage.

3) In Fig.3f, it appears that there are still a significant percentage of CD31+ cells present in MECOM KO cells (19.4%). If MECOM is required for endothelial cell differentiation, then what accounts for the CD31 expression? Is this because CD31 expression is initiated independent of MECOM?

4) Does knocking out MECOM by CRISPR in HUVECs affect the proliferation/survival of these cells? It

appears so from the data provided in Fig. S5b, as well as from the field of view provided in Fig.4i. All the downstream effects observed in MECOM KO cells might be a result of defective cell viability. The authors should address this. This is also relevant since in the RNA seq experiments performed in HUVEC KO (?) cells, the top GO term for down regulated genes is "regulation of cell proliferation".

4a) Additionally, do the HUVECs still maintain their endothelial identity (based on marker gene expression) upon knocking out MECOM?

5) In Fig.5, for all the zebrafish experiments, the authors should clearly mention either in the results or the methods whether the MECOM CRISPR KO analysis was performed as an F0 analysis or whether stable lines were established, and knockouts were obtained by intercrossing heterozygous animals. This is important in terms of the controls used especially if the experiments were performed as F0 analyses, for which scrambled guides are appropriate controls and not just wild type embryos. If stable KO lines were established, does the loss of MECOM result in embryonic lethality?

5a) Further are MECOM RNA/protein levels reduced in MECOM KO animals? The authors need to establish whether this is a knockout or not?

5b) The authors show that MECOM CRISPR KO in zebrafish leads to impaired vasculogenesis. Previous studies have shown that morpholino knockdown of MECOM in zebrafish does not cause aberrant vascular development. The authors should discuss their results in light of this previous data.

Minor comments:

6) Higher resolution images with better contrast are required for Fig.4e. It is difficult to make any sort of assessment of the wound closure phenotype in the scratch assay.

7) For RNA seq data obtained in Fig.6, the authors should clearly mention either in their results or methods how MECOM expression was abrogated. The authors appear to use the terms knockdown and knockout interchangeably for this part of the manuscript, which might have different implications on the use of controls as well as how the data are interpreted.

8) For Fig.5d, the gfp transgene (Fli1:gfp) should be denoted on the figure panel and/or in the figure legend.

9) Lines 249-250, the authors should reference Fig. S2f in the text.

10) The Discussion section largely reiterates the results. It would be useful to use this section to place the results of the present study into context, particularly with regard to previously published studies in mice and in hematopoietic system, described in comments above.

Response Letter

We would like to thank the Reviewers for having taken in reviewing our manuscript thoughtfully. We appreciate the reviewers for acknowledging the significance and novelty of our findings, and are grateful for the Reviewer's suggestions. By meticulously responding to their guidance, the manuscript has been substantially improved. We are optimistic that the manuscript is now suitable for publication in Nature communications.

Response to Reviewer #1:

Lv et al present a manuscript in which they identify MECOM as regulator of endothelial cell differentiation. This is done by leveraging a large number of ChIPseq datasets in HUVECs (human umbilical vein endothelial cells). From these difficult to grasp analyses MECOM is found as a key regulator. The authors go on to show using functional experiments that MECOM indeed is a functional regulator of endothelial differentiation. The generate knock-outs of MECOM in human iPSCs, which fail to differentiate to endothelial cells. In zebrafish knock-out of mecom results in impaired formation of vasculature. Finally, the authors show that MECOM regulates VEGFR2 through a distal regulatory element. Deletion of this element prevents the upregulation VEGFR2 in HUVEC. Overexpression of VEGFR2 in mecom knock-out zebrafish can rescue the impaired vasculature, suggesting that MECOM regulates endothelial differentiation through VEGFR2.

A novel regulatory role is presented for MECOM in endothelial differentiation. The functional experiments are very nice and presented in a clear manner. The rationalization for MECOM based on the ChIPseq data difficult to follow and not the most exciting part of the manuscript.

Response: We are grateful to the reviewer for the many positive comments in this very thoughtful summary. For instances, the reviewer appreciated that the regulatory role of MECOM in EC is "novel", and the functional experiments are "very nice and presented in a clear manner". We thank the reviewer for reminding us that the rationalization for MECOM based on the ChIPseq data is difficult to follow, likely because of the lack of sufficient details. We therefore added many important details in the manuscript and substantially revised the Methods section (see point-by-point responses below). We also moved some of the associated bioinformatics figures to supplemental figures, so that the main figures will be more concise and clearer. We hope this substantial improvement lets readers feel more focused on the regulatory role of MECOM in endothelial cells, thus feel more excited about the major discovery.

1. I have one major concern with the manuscript. The authors claim that MECOM target sites form loops to regulate target genes. The wording is already very suggestive, because this is all based on correlation. I suggest the authors delete the distal MECOM binding site at the NR2F2 locus (Fig. 6l) and perform a 4C analysis to see whether loss of this binding site leads to a change in looping and gene expression.

Response: We are grateful to the reviewer for this constructive suggestion. Following this suggestion, we first analyzed the MECOM Chip-seq data and binding motif. We identified the MECOM binding site in the enhancer that is connected to NR2F2 promoter by the chromatin loop (**Fig. 5i, j**). Using CRISPR-cas9 system, we deleted this binding site using two pairs of gRNA sequences around this region (**Fig. 5j top**). We next performed chromosome conformation capture (3C) assays based on 5 pairs of primers (**Fig. 5j bottom**, 3C primer design was described in SUPPLEMENTAL METHODS). We observed that the interaction frequency between the enhancer and promoter of NR2F2 was significantly reduced by the deletion of the MECOM binding site (**Fig. 5k**). We also knocked down MECOM expression in

HUVEC and then performed 3C assays. The data showed that the interaction frequency between the enhancer and promoter of NR2F2 was significantly reduced by the knockdown of MECOM (**Fig. 5l**). Further, The RNA expression level of NR2F2 was also significantly downregulated after we deleted the binding site (**Fig. 5o**) and knocked down MECOM (**Fig. 5p**). These results verified that MECOM expression and the MECOM target site is required to form the chromatin loop and activate the transcription of NR2F2.

Other points:

2. The authors describe MECOM as a master regulator and it is clear that loss of this factor impairs the endothelial development, however, loss of MECOM can also be rescued by the overexpression of one factor. Can it be justified to call this a master regulator?

Response: We thank the reviewer for this comment. We described MECOM as a master regulator because MECOM is connected by the largest number of network edges to other EC regulators (**Fig. 1i, Supplementary Table 3**). Notably, the VEGFR2 that is activated by MECOM and can rescue the effect of MECOM loss is a signaling receptor well known to play a pivotal role in endothelial cell identity regulation. It is the key receptor of the vascular endothelial growth factor (VEGF) signaling pathway, which further regulate other critical downstream endothelial pathways such as the endothelial proliferation, migration, adhesion, etc. VEGF is an extracellular factor frequently used both to induce endothelial differentiation and culture endothelial cells. Further, VEGF has been used in many preclinical animal models and multiple clinical trials to target angiogenesis (Rohner, E., et al. Unlocking the promise of mRNA therapeutics. *Nat Biotechnol.* 2022). Our knockout and rescue experiments demonstrated that VEGF signaling is a direct downstream pathway activated by MECOM in endothelial cells, thus suggested a key role of MECOM in regulating endothelial identity. However, we agree with the reviewer that it might be a little better to be more conservative and thus describe MECOM as an EC “lineage regulator” instead of “master regulator”. We have made corrections accordingly throughout the manuscript.

3.1 In the MECOM KO iPSCs OCT4 and SOX2 remain expressed. Is pluripotency maintained? Can the authors do classical analyses to measure pluripotency (for instance alkaline phosphatase staining)?

Response: As MECOM express significantly lower in iPSCs and other cell types than in endothelial cells (**Fig. 1e-j, S3c**), we speculate that MECOM KO is likely to impair endothelial differentiation rather than iPSC pluripotency. To answer this important question, we performed alkaline phosphatase (AP) staining in control and MECOM KO iPSC. We found that the AP activity level didn't show significant differences between WT and MECOM KO iPSC (**Fig. S5h, i**). We further performed qPCR of iPSC pluripotency markers in WT and MECOM KO iPSC. We found that MECOM KO did not decrease the expression of pluripotency marker genes in iPSC, including the OCT4, SOX2, NANOG, TERT, ZFP42, and UTF1 (**Fig. S5g**). We instead observed either significant or slight up regulation of these marker genes in iPSCs upon MECOM KO. Those data are consistent with our conclusion that MECOM depletion significantly reduced the yield of ECs after differentiation, without impairing iPSC pluripotency.

We further performed AP staining in the differentiated cells derived from control and MECOM KO iPSC and found these cells all show negative AP staining. Therefore, although failed to differentiate towards the endothelial lineage and express higher level of iPSC marker genes in the MECOM KO cells when compared to the wild type cells, these cells all have lost pluripotency to an undetectable degree based on AP staining. One possibility, however, is that the MECOM KO cells were differentiating towards some unknown lineages that retain certain

degree of OCT4 and SOX2 expression but lose alkaline phosphatase (for which the underlying mechanism could be beyond the scope of the current manuscript).

3.2 Related to this: it is strange that the OCT4 and SOX2 levels are normalized to the KO instead of the control.

Response: In control iPSC-derived EC, the gene expression level of SOX2 is undetectable, thus we couldn't calculate differential fold change between the control and MECOM KO group. To show this big difference, we labeled the expression of SOX2 as "1" in the MECOM KO group and "UD" (undetectable) in the control group. The gene expression level of OCT4 in the control group is also very low (close to 0), thus was presented in a similar way for SOX2.

4. Figure 1h: p-values are not a measure for enrichment, a low p-value does not denote strong enrichment, but could just mean a large sample size. Please plot the enrichment scores instead of p-values.

Response: The previous "figure 1h" is now "figure 1a". The reviewer is right that typically a p value can be more significant when the fold enrichment is the same while sample size is larger. Therefore, we have paid attention to this effect by using the same number of genes (1000 genes) for each group defined by the epigenetic features to calculate the P value. In this way, the difference in these P values will no longer be associated with the analyzed gene number (sample size).

Also, in this analysis, because the very top genes (e.g., ranked by H3K4me3 width) tend to be more reliable, other enrichment score such as the fold enrichment can also change with the number of analyzed top genes (sample size). Therefore, we reasoned that using the same number of top genes for the analysis is a better solution to avoid the effect of sample size.

However, we noted that the reviewer's suggestion is further helpful here because showing enrichment scores in addition to the P values will make the result more meaningful. Therefore, we have added figures to show enrichment scores including the fold of enrichment and the absolute number of overlapped genes (**Fig. S2e-f**). The P values, fold enrichment, and absolute number of overlapped genes show consistent observations, i.e., the top genes ranked by the histone modifications H3K4me1, H3K4me2, H3K9ac, H3K27ac and H3K4me3 displayed the strongest enrichment of the EC positive regulators when compared to the other histone modifications. A clearer difference in the enrichment is observed when we use P value rather than the fold enrichment or absolute gene number, suggesting that the P value is a better score to distinguish between the usefulness of these histone modifications as signatures of cell identity genes.

**I. 166: the authors talk about hub genes and gene regulation network, but it is completely unclear where this comes from.*

Response: Thanks so much for the comment. To make it clearer, we included a "Gene Regulatory Network (GRN) analysis" section in the Supplemental Methods to address this concern in detail. We further added "We next evaluated the importance of these 1000 top-ranked genes in EC gene regulation network constructed using the CellNet algorithm (See Gene Regulation Network analysis in Supplemental Method)" in the related section of the manuscript.

**l. 171: "our method", what is our method, this is not clearly defined.*

Response: We thank the reviewer for this comment. It is the method described at the beginning of that paragraph. Motivated by this helpful comment, we added "i.e., ranked by the average rank of widths of the five histone modifications" following the "our method", so that it can let readers feel easier to connect it to the method described at the beginning of that paragraph.

**ref. 25: please cite the original research paper, and not only a preview paper.*

Response: We thank the reviewer for this important reminder, and have cited the following paper:
Hsieh, T. S., et al. (2020). "Resolving the 3D Landscape of Transcription-Linked Mammalian Chromatin Folding." *Mol Cell* **78**(3): 539-553 e538.

**Figure 6c is unreadable*

Response: The previous "figure 6c" is now "figure 5c". We guess that the reviewer refers to a format problem in the table. We have made corrections accordingly.

**Figure 7r: there is no 7r in the figure set.*

Response: It was supposed to be **Fig. 7q (Fig. 6q in the new manuscript)**. We have made corrections accordingly.

Response to Reviewer #2:

In this manuscript, Lv et al leverage available epigenomic and transcriptomic datasets to curate and identify novel regulators of endothelial identity. From this effort, the authors identify MECOM as an endothelial restricted transcription factor with several important functions in endothelial development and function. The authors show that loss of MECOM function results in impaired endothelial development in fish, defective endothelial cell differentiation from iPSCs, and disrupted endothelial cell functions in cultured endothelial cells. Additionally, the authors identify many endothelial genes and regulatory elements downstream of MECOM, including numerous components of the VEGF signaling pathway, and they infer the MECOM binding site from existing ChIP-seq datasets.

The identification of MECOM as a regulator of endothelial cell specification and differentiation would be an important advance and if confirmed to be true (see below) will be an important contribution to the field. Moreover, the approach of using existing datasets to identify cell specific regulators, particularly leveraging epigenomic datasets, is important and creative, and I believe would appeal to readers. Thus, in principle, I think this work could be suitable for publication in Nature Communications. However, there are several very significant issues, both technical and with regard to the existing published literature that affect the novelty and interpretation of the current work that absolutely should be fully addressed prior to any consideration for publication.

Response: We are grateful to the reviewer for the many positive comments in this very thoughtful summary. We also thank the reviewer for raising several issues, which we found to be very constructive and addressed carefully in the point-to-point responses below.

1) The authors primarily use data generated from HUVECs to identify MECOM as an upregulated transcription factor in endothelial cells and all comparisons are done mainly with respect to differential signatures of chromatin marks in ES cells. However, it is well known that MECOM is highly expressed in hematopoietic stem cells. It is also widely established that hematopoietic and endothelial cells share a common progenitor and are closely related cell types. This is not addressed in the manuscript.

Response: Please see responses to 1a) and 1b) below.

1a) Moreover, the authors should compare MECOM expression in hematopoietic progenitors as well as chromatin signatures in these cells would help clarify enrichment levels on MECOM in endothelial cells in comparison to its expression level and signatures in known cell types.

Response: We are grateful to the reviewer for providing this helpful suggestion. We investigated the H3K4me3 signal around the *MECOM* locus between HPC and HUVEC and observed comparable H3K4me3 peak width between the two cell types (**Fig. 1j**). Therefore, we added two paragraphs in the manuscript (the 2nd and 4th paragraphs of the Discussion section) to discuss related literature, which reported that MECOM is required for the formation of HPC but inhibits the differentiation of HPC towards downstream lineages (immune cells). To further check the chromatin status around MECOM in such cell types, we visualized DNase-Seq signal (which reflects chromatin accessibility) around MECOM locus in HUVEC, HPC, Myeloid Progenitor Cell, and several immune cell types (**Fig. S3b**). Consistent with the observation for H3K4me3 (**Fig. 1j**), MECOM locus had the strongest chromatin openness (more peaks and broader signal) in endothelial cells, including both HUVEC and HPAEC, whereas MECOM in HPC was associated with the second strongest chromatin openness compared to other cell types (**Fig. S3b**). Besides chromatin openness, following the reviewer's suggestion, we also checked MECOM mRNA abundance in these cell types. We observed that MECOM was highly expressed in the different types of endothelial cells, but less highly expressed in HPC and lowly expressed in immune cells (**Fig. S3c**). These chromatin state and expression data is consistent with our discovery of an important role of MECOM in the endothelial lineage and the known lineage relationship between endothelial cells and HSPC.

1b) In Fig.3a-d, the authors analyze scRNA seq data from Day 8 and Day 12 of iPSC to EC differentiation and show that MECOM is expressed in Cluster 0, which they identify as endothelial cluster based on marker identity. However, most of the markers used to determine endothelial identity are also expressed in hematopoietic cells. The authors need to show that this cluster does not express other hematopoietic progenitor markers such as Runx1, CD41 or CD45 and is truly an endothelial cluster.

Response: We thank the reviewer for this important suggestion. We thus checked the HPC markers and systematically assessed the enrichment of HPC differentiation genes in Cluster 0.

We showed in the manuscript that MECOM and VEGFR2 were highly expressed in Cluster 0 (**Fig. 2a, b**), which appears to be an endothelial cell cluster. We now further checked the expression of CDH5 (**Fig. S4a**), a well-known endothelial cell marker, and the HPC markers including ITGA2B (CD41) and RUNX1 (**Fig. S4b-c**) suggested by the reviewer. The result showed that the expression of endothelial marker CDH5 was highly enriched in Cluster 0 relative to other clusters. However, the HPC markers CD41 and RUNX1 were very lowly expressed across all the clusters. We did not observe any expression of CD45 in this dataset.

We next directly compared the expression of HPC and EC identity genes in cluster 0. To do this, we collected a set of HPC differentiation genes from Gene Ontology (GO:0002244), and calculated their average expression across cells in Cluster 0 in comparison to the set of EC positive regulator genes (described in the manuscript, Supplementary Table 2). Cluster 0 cells expressed significantly higher level of EC identity genes compared to HPC differentiation genes (**Fig. S4d**). These results indicated that the cells in Cluster 0 would be endothelial cells rather than HPC.

1c) If there are scRNAseq data available for earlier timepoints of this experiment (day 0 and day 4), it would be useful to show when MECOM expression begins and whether it precedes the expression of other endothelial markers.

Response: Following the reviewer's suggestion, we found a public scRNA-seq dataset (GSE131736, Ian R McCracken, et al., 2020, European Heart Journal) that covers cells on day 0, 4, 6, 8, and 12 from iPSC to EC differentiation. In this dataset, MECOM and other EC marker genes including KDR (VEGFR2), PECAM1 (CD31), and CDH5 (CD144) were not expressed until day 6 after differentiation. We were not able to evaluate whether the MECOM expression precedes the other EC markers (**Fig. S4e**). This might be due to the unavailability of smaller time window. In addition, drop-out issue of scRNA-Seq might make it hard to detect the low expression of these genes at early stage of endothelial differentiation. MECOM expression level might be low at the start point of MECOM expression during the differentiation, and thus might not be detectable by scRNA-Seq. Therefore, we carried out qPCR to detect MECOM expression on each day of iPSC-EC differentiation from day 0 to day 6. We found that MECOM started expressing on day 2 in the differentiation process, and its expression upregulation was greater than that of several tested EC markers (CDH5, KDR, DLL4, and EFNB2) on days 2 and 3, whereas the upregulation of CDH5 becomes greater after days 4 (**Fig. S4f**). This result suggested that MECOM expression preceded the expression of these known EC marker genes in the differentiation process.

2a) Previous studies have shown that loss of MECOM in mice, embryonically as well as postnatally, primarily affects hematopoiesis with little or no effect on endothelial development. Additionally, conditional loss of MECOM in the endothelial lineage using Tie2-Cre results mainly in hematopoietic defects (not endothelial defects), suggesting that MECOM is not essential to maintain endothelial identity. The authors should discuss these published studies in greater detail and reconcile their data with these previously published papers.

Response: We thank the reviewer for this constructive comment. As an important transcription factor, MECOM plays an essential role in regulating gene expression during early development and hematopoiesis. The role of MECOM in regulating heart and vessel development was reported in some of these publications, although the role of MECOM in endothelial cells was not the focus of these publications and thus not yet clear. Following the reviewer's suggestions, we discussed and cited related publications in the manuscript:

1. Dr. Kurokawa's group¹ reported that EVI1 (MECOM) is a critical regulator for hematopoietic stem cells (HSC) and transformed leukemic cells. They generated EVI1 mutant mice and observed defective HSC activity in EVI1-deficient embryos. They further selectively disrupted EVI1 function in Tie2+ endothelial and hematopoietic stem/progenitor cells, and observed a similar defect that mimics the Evi-1 deficiency. This paper mainly focused on investigating the role of EVI1 in hematopoiesis, thus didn't pay much attention to observing defects in the heart and vessels. However, the author did mention that "After E13.5, many EVI1-/- embryos exhibited.....defective large vessel development" (Please see the Figure 1F of the publication of Goyama, et al, *Cell Stem Cell*, 2008)¹.

2. An earlier paper (*Mech Dev*, 1997)² also reported that EVI1 mutant mice showed severe heart malformations. Here are some summaries of their observations:

- 1) By 10.5 d.p.c., EVI1mutants hearts were clearly falling. Gross observations revealed that general circulation was poor, and that the beating heart produced only a refluxing of blood.
- 2) The improper heart development is manifested in a looping defect giving the heart a more tube-like appearance.
- 3) EVI1 mutants display a poorly developed constriction, and a definite looping defect that resembles retarded or arrested heart development.
- 4) Histological sections revealed that the degree of chamber development was delayed in EVI1 mutant embryos. A consistent decrease in trabeculations was evident by 9.5 d.p.c.
- 5) The failed heart is indicated also by blood and fluid in the pericardial sac, and indication of pericardial effusion.

3. In 2014, Dr. Copeland's group³ further reported that mice carrying a hypomorphic EVI1 allele exhibited severe congenital heart defects. Here are some summaries of their observations:

- 1) Common arterial trunk (CAT), where two great arteries fail to separate and leave the heart as one common vessel, was observed in 3 out of 6 Evi1mutant embryos.
- 2) Double outlet right ventricle (DORV), where both the aorta and pulmonary trunk leave one ventricle, was observed in half of the Evi1 mutant embryos.
- 3) Aortic arch formation impairments were found in 4 out of 6 Evi1mutant embryos.
- 4) These types of congenital heart defects.....represent the major cause of the perinatal lethality seen in Evi1mutant pups.

Considering the critical role of endothelial cells in heart tube, chamber, and vessel formation⁴, the observations in these publications might be consistent with our observation of the role of MECOM in regulating endothelial cell function and lineage specification. We included and discussed those published studies in the second paragraph of the Discussion section of the manuscript.

2b) Given the lack of requirement for MECOM for endothelial cell development in mice, the authors may be overstating the role of MECOM as “a master regulator” of the endothelial lineage.

Response: Thanks so much for the comments. According to our responses to question 2a), MECOM plays an important role in heart and vessel development. In the current manuscript, we reported the role of MECOM in maintaining the endothelial cell identity. We described MECOM as a master regulator because MECOM is connected by the largest number of network edges to other EC regulators (**Fig. 1i, Supplementary Table 3**). However, we agree with the reviewer's suggestion to be more conservative with our statement. Therefore, we will describe MECOM as an EC “lineage regulator”, instead of “master regulator”. We have made corrections accordingly throughout the manuscript.

3) In Fig.3f, it appears that there are still a significant percentage of CD31+ cells present in MECOM KO cells (19.4%). If MECOM is required for endothelial cell differentiation, then what accounts for the CD31 expression? Is this because CD31 expression is initiated independent of MECOM?

Response: The previous “figure 3f” is now the “figure 2f”. We hypothesized that such CD31+ CDH5- cells were probably not endothelial cells. To address this question, we checked the specificity of CD31 in different cell types using RNA-Seq data. As expected, *PECAM1* (CD31) was highly expressed in different types of endothelial cells, however, it also showed a high or

moderate expression level in macrophage, HPC, and immune cells (**Fig. R1a**). In contrast, *CDH5* (CD144), another endothelium marker frequently used in literature, was expressed more specifically in endothelial cell compared to HPC and immune cells (**Fig. R1b**). This indicated that CD31 alone may not be an ideal marker for characterizing endothelial cells. Instead, it is more reasonable to use the combination of CD31 and CDH5 to sort the fraction of endothelial cells. Therefore, we reasoned that those 19.4% CD31+ and CDH5- cells were not endothelial cells and maybe other uncharacterized cell types.

We observed that CD31 mRNA level was slightly reduced upon MECOM knockout in HUVEC, but the reduction was not to a significant magnitude (**Fig. S6a**). Based on the ChIP-seq data and motif analysis (**Fig. R1c**), although a few MECOM motifs were detected (red track), we failed to observe ChIP-Seq enrichment around *PECAM1* gene locus when compared to Input sample. However, we found much more MECOM motifs around *CDH5* gene locus as well as strong ChIP-Seq enrichment on the proximal upstream of *CDH5* gene (**Fig. R1d**). Thus, it is likely that MECOM does not directly regulate *PECAM1* but directly regulates *CDH5* gene expression, although it is possible that MECOM might regulate *PECAM1* indirectly through unknown mechanism. These results are consistent with the observation that CD31 was not significantly reduced upon MECOM knockout. As we discussed above, *CDH5* is a more specific marker for endothelial cells compared to CD31, the significant reduction of *CDH5* but not CD31 under MECOM perturbation further supports the conclusion that MECOM is more likely a regulator of endothelial lineage rather than the other lineages that also express CD31.

4a.1) Does knocking out MECOM by CRISPR in HUVECs affect the proliferation/survival of these cells? It appears so from the data provided in Fig. S5b, as well as from the field of view provided in Fig. 4i.

Response: The reviewer is right. Our results from the CellTiter-Glo Luminescent Cell Viability Assay, which is an assay for measuring cell proliferation and viability, indicated a significant effect of MECOM KO (**Fig. 3i, which is the previous Fig. S5b**) as well as

MECOM KD (**Fig. R2**) in HUVECs. This is further consistent with our RNA-Seq analysis result, which indicates significant enrichment of the EC proliferation pathway (GO:0001936) in the genes down-regulated by MECOM KO in HUVECs (**Fig. 5b**)

4a.2) All the downstream effects observed in MECOM KO cells might be a result of defective cell viability. The authors should address this. This is also relevant since in the RNA seq experiments performed in HUVEC KO (?) cells, the top GO term for down regulated genes is “regulation of cell proliferation”.

Response: Thanks for this important suggestion. To test if the downstream effects observed in MECOM KO cells were induced by defective cell proliferation/viability, we treated the WT and MECOM KO HUVEC with 5ug/ml Mitomycin C, which has been reported to significantly inhibit cell proliferation. By performing cell viability assay at 24h and 48h after Mitomycin C treatment, we found that the HUVEC in WT and MECOM KO groups both stop proliferation and thus stay

quiescent. (**Fig. S7a**). We next performed endothelial function assay (tube formation, NO production, LDL uptake, and cell migration) in WT and MECOM KO HUVEC treated with Mitomycin C. Consistent with our previous results, we found that MECOM disruption still significantly impaired endothelial functions, as indicated by reduced tube formation in vitro, migration, LDL uptake, and nitric oxide production (**Fig. S7b-i**). Further, these observed effects were reproducible when we performed knockdown of MECOM in HUVECs (**Fig. R3**). Based on those data, we could make the conclusion that the impaired EC function was induced by MECOM KO, and is not simply caused by defective cell proliferation/viability.

4b) Additionally, do the HUVECs still maintain their endothelial identity (based on marker gene expression) upon knocking out MECOM?

Response: Our result indicates that the HUVEC lost endothelial identity upon MECOM KO. Following the reviewer's comments, we checked the expression changes upon MECOM knock out (KO) for well-known EC marker genes, e.g., the CDH5 and VEGFR2. MECOM KO significantly reduced CDH5 and VEGFR2 mRNA levels in HUVEC cells (**Fig. S6a**). Furthermore, we investigated whether this reduction happened on most EC identity genes upon MECOM depletion in HUVEC cells. We conducted a Gene Set Enrichment Analysis (GSEA) for the differential expression of EC positive regulator genes (collected as described in the manuscript, Supplementary Table 2) in MECOM KO versus CT cells. The result showed that EC positive regulator genes were significantly enriched with down-regulated genes upon MECOM KO (**Fig. S6b**), suggesting loss of endothelial identity upon knocking out MECOM. This is consistent with the loss of typical endothelial functions upon MECOM KO in HUVEC (**Fig. 3**).

5) In Fig.5, for all the zebrafish experiments, the authors should clearly mention either in the results or the methods whether the MECOM CRISPR KO analysis was performed as an F0 analysis or whether stable lines were established, and knockouts were obtained by intercrossing heterozygous animals. This is important in terms of the controls used especially if the experiments were performed as F0 analyses, for which scrambled guides are appropriate controls and not just wild type embryos. If stable KO lines were established, does the loss of MECOM result in embryonic lethality?

Response: Thanks so much for the helpful comments and important question. All the MECOM CRISPR/Cas9 KO zebrafish studies were performed in F2 offspring which were obtained from F1 inbreeding. We added details in the Methods and pasted here for reference:

“Genotyping of germline mutants

To assess germline inheritance of genome modification generated in somatic mutant fish, P0 founders from MECOM sgRNA injection experiment were outcrossed to WT zebrafish and genomic DNA was extracted from F1 larvae or tailfin clips of adult F1 fish. DNA from individual fish was used as a template for subsequent PCR using primers spanning the target site of MECOM sgRNA. PCR amplicons encompassing the target region were analyzed by Sanger sequencing to detect sequence variants. F2 larvae were obtained from F1 inbreeding and used for all the experiments in this study. The same primers were used for PCR amplification of genomic DNA in order to discriminate WT, heterozygous and homozygous carriers of the MECOM allele among the offspring of subsequent inbreeding experiments.”

The stable MECOM KO fish line was established, and we didn't observe embryonic lethality. The homozygous MECOM mutants were successfully bred into adult fish in consistent with previous report (Shull, Lomeli Carpio, et al. The conserved and divergent roles of Prdm3 and Prdm16 in zebrafish and mouse craniofacial development. *Developmental biology* 461.2 (2020): 132-144.)

5a) Further are MECOM RNA/protein levels reduced in MECOM KO animals? The authors need to establish whether this is a knockout or not?

Response: We included new data to confirm the deficient RNA and protein expression of MECOM in knockout mutants (**Fig. 4d-e**).

5b) The authors show that MECOM CRISPR KO in zebrafish leads to impaired vasculogenesis. Previous studies have shown that morpholino knockdown of MECOM in zebrafish does not cause aberrant vascular development. The authors should discuss their results in light of this previous data.

Response: As the reviewer noted, in the prior studies (<https://www.ncbi.nlm.nih.gov/pmc/articles/PMC5090218>), MECOM knockdown does not affect the establishment of dorsal aorta or the major vein – a process known as vasculogenesis. However, this paper did not show the phenotype of the intersegmental vessels (ISV), which is regulated by angiogenesis. In our studies, we found that indeed MECOM knockdown using MO or knockout using CRISPR/Cas9 has no effect on vasculogenesis, which is consistent with the prior study. On the other hand, our data reveal that MECOM deficiency impairs angiogenesis – the development of ISV. We have now added the following discussion of our results to the new manuscript: “Consistent with previous report, MECOM KO or knock down didn’t affect the establishment of dorsal aorta or the major vein, a process known as vasculogenesis. Instead, our data showed that MECOM deficiency impairs angiogenesis, which includes the establishment of ISV and SIV.”

Minor comments:

6) Higher resolution images with better contrast are required for Fig.4e. It is difficult to make any sort of assessment of the wound closure phenotype in the scratch assay.

Response: We thank the reviewer for the suggestion. We have revised the figure accordingly. The previous “figure 4e” is now “figure 3e”.

7) For RNA seq data obtained in Fig.6, the authors should clearly mention either in their results or methods how MECOM expression was abrogated. The authors appear to use the terms knockdown and knockout interchangeably for this part of the manuscript, which might have different implications on the use of controls as well as how the data are interpreted.

Response: We appreciate the reviewer for this important suggestion. It is the pool of cells treated with MECOM CRISPR-Cas9. To distinguish this knockout pool from knockout single clone, we revised the manuscript to describe it as “MECOM knockout cell pool”, and use “MECOM knockout single clone” when we describe a single clone.

8) For Fig.5d, the gfp transgene (Flil:gfp) should be denoted on the figure panel and/or in the figure legend.

Response: We revised the figure accordingly.

9) Lines 249-250, the authors should reference Fig. S2f in the text.

Response: Thanks. We have updated the description. The old Fig. S2f has been updated and is now the new **Fig. S3h**.

10) *The Discussion section largely reiterates the results. It would be useful to use this section to place the results of the present study into context, particularly with regard to previously published studies in mice and in hematopoietic system, described in comments above.*

Response: Thanks so much for this very useful suggestion. The new manuscript now included and discussed the previously published studies of MECOM mutant mice and their impaired function in hematopoietic system (the second and fourth paragraphs in the Discussion section).

REFERENCE

1. Goyama S, Yamamoto G, Shimabe M, Sato T, Ichikawa M, Ogawa S, Chiba S, Kurokawa M. Evi-1 is a critical regulator for hematopoietic stem cells and transformed leukemic cells. *Cell Stem Cell*. 2008; 3:207-20.
2. Hoyt PR, Bartholomew C, Davis AJ, Yutzey K, Gamer LW, Potter SS, Ihle JN, Mucenski ML. The Evi1 proto-oncogene is required at midgestation for neural, heart, and paraxial mesenchyme development. *Mech Dev*. 1997; 65:55-70.
3. Bard-Chapeau EA, Szumska D, Jacob B, Chua BQ, Chatterjee GC, Zhang Y, Ward JM, Urun F, Kinameri E, Vincent SD, Ahmed S, Bhattacharya S, Osato M, Perkins AS, Moore AW, Jenkins NA, Copeland NG. Mice carrying a hypomorphic Evi1 allele are embryonic viable but exhibit severe congenital heart defects. *PLoS One*. 2014; 9:e89397.
4. Kim H, Wang M, Paik DT. Endothelial-Myocardial Angiocrine Signaling in Heart Development. *Front Cell Dev Biol*. 2021; 9:697130.

REVIEWERS' COMMENTS

Reviewer #1 (Remarks to the Author):

Lv et al have addressed most of my comments. The manuscript is substantially improved compared to the original version.

However, I suggested the authors do a 4C experiment to test whether the distal enhancer of NR2F2. Instead, they performed a 3C experiment. One of the difficulties with 3C is that it can be very unreliable [see for instance <https://pubmed.ncbi.nlm.nih.gov/16369547/> by Job Dekker]. With 4C these problems are largely mitigated and gives an idea how the interactions between the promoter are changed.

Reviewer #2 (Remarks to the Author):

The authors have fully addressed all of my concerns. Indeed, the authors should be commended for the very thorough and thoughtful revision and rebuttal. I support publication in Nature Communications.

Response Letter

We would like to thank the Reviewers for thoughtfully reviewing our revised manuscript. We are very glad to know that we have addressed most of the comments and suggestions from the reviewers. By following the suggestion from reviewer 1, we further improved our manuscript by discussing the limitations of 3C in the results section. We are optimistic that the manuscript is now suitable for publication in Nature communications.

Response to Reviewer #1:

Reviewer #1 (Remarks to the Author):

Lv et al have addressed most of my comments. The manuscript is substantially improved compared to the original version.

However, I suggested the authors do a 4C experiment to test whether the distal enhancer of NR2F2. Instead, they performed a 3C experiment. One of the difficulties with 3C is that it can be very unreliable [see for instance <https://pubmed.ncbi.nlm.nih.gov/16369547/> by Job Dekker]. With 4C these problems are largely mitigated and gives an idea how the interactions between the promoter are changed.

Response: We are grateful to the reviewers for reviewing our revised manuscript. We are thrilled to hear that we *have addressed most of the comments and the manuscript is substantially improved*. We agree with the reviewer that 3C experiment has its limitation, as primer annealing efficiency can potentially bias the contact frequency. To mitigate this issue, we designed and tested five pairs of primers, which yielded highly consistent results. Additionally, we have taken the comment into careful consideration and added several sentences to the results section to provide a more objective discussion of the data, including the limitations of the 3C experiment and potential ways to further enhance the observation in future.

The sentences we added are as follows:

“We noted that 3C experiment might have its limitation, as primer annealing efficiency can potentially bias the contact frequency. Therefore, we used 5 pairs of primers to mitigate this bias. Other comprehensive and high-resolution methods such as 4C or Hi-C might be helpful to further validate the observation in future.”

Response to Reviewer #2:

Reviewer #2 (Remarks to the Author):

The authors have fully addressed all of my concerns. Indeed, the authors should be commended for the very thorough and thoughtful revision and rebuttal. I support publication in Nature Communications.

Response: We are grateful to the reviewer for reviewing the revised manuscript. We appreciate the very positive kind words and thank the reviewer for *commending for the very thorough and thoughtful revision and rebuttal and supporting publication in Nature Communications*.